# Global distribution of surface soil organic carbon in urban greenspaces

Hongbo Guo [1,2], Enzai Du [1,2] ✉, César Terrer [3] & Robert B. Jackson [4,5]

Urban greenspaces continue to grow with global urbanization. The global distribution and stock of soil organic carbon (SOC) in urban greenspaces remain largely undescribed and missing in global carbon (C) budgets. Here, we synthesize data of 420 observations from 257 cities in 52 countries to evaluate the global pattern of surface SOC density (0–20 cm depth) in urban greenspaces. Surface SOC density in urban greenspaces increases significantly at higher latitudes and decreases significantly with higher mean annual temperature, stronger temperature and precipitation seasonality, as well as lower urban greenness index. By mapping surface SOC density using a random forest model, we estimate an average SOC density of 55.2 (51.9–58.6) Mg C ha$^{-1}$ and a SOC stock of 1.46 (1.37–1.54) Pg C in global urban greenspaces. Our findings present a comprehensive assessment of SOC in global urban greenspaces and provide a baseline for future urban soil C assessment under continuing urbanization.

Urban areas cover 0.3–0.6% of the world's land[1,2] but account for more than 70% of global energy-related $CO_2$ emissions[3]. With continuing urban expansion, urban areas are becoming increasingly important in the global carbon (C) cycle[4]. In parallel, urban greenspaces (e.g., urban forests and lawns from parks, residential areas, and gardens) are growing rapidly and store increasing amounts of C in vegetation and soils[5]. Soil organic carbon (SOC) is essential for urban soil nutrient retention that both benefits plant growth and alleviates nutrient leaching to urban streams[6,7]. Although an increasing number of studies have reported SOC densities (i.e., the amount of SOC to a certain depth per unit area) in urban greenspaces for individual cities[8,9] or nationally[10,11], the global distribution, controlling factors, and magnitude of SOC stocks in urban greenspaces remain poorly characterized[12]. A quantitative analysis of SOC density and stocks in global urban greenspaces is needed to fill this gap.

Global patterns of SOC density in natural terrestrial ecosystems have been well documented[13–15]. In urban greenspaces, some early studies demonstrated a human-induced spatial convergence of urban soil properties among cities because of strong anthropogenic interventions and relatively similar management operations[16–18]. At a much broader scale, recent observational analyses have suggested that there is nevertheless a strong latitudinal pattern and climate control of urban SOC density[19].

Theoretically, SOC density in urban greenspaces is co-regulated by natural and anthropogenic drivers[20]. Urban greenspace SOC density can be affected by a city's climate as well as potential legacy effects of pre-urban land use[21]. The urban heat-island effect may increase SOC loss by accelerating the decomposition of soil organic matter[20]. Vegetation conditions (e.g., vegetation type and coverage) and horticultural management (e.g., fertilization and irrigation) can also affect the spatial variation of SOC in urban greenspaces at large scales[22] by modifying vegetation growth and soil biogeochemical cycles[6,23]. However, the relative importance of the potential drivers of SOC density in urban greenspaces remains to our knowledge unquantified globally.

Similarly, despite rapid urbanization and consequent expansion of urban greenspaces, the total SOC stocks in global urban greenspaces remain unquantified. Several studies have reported SOC stocks of urban greenspaces at city or national scales[10,24], primarily in relatively wealthy countries[25,26] (e.g., USA and UK). However, such studies

[1]State Key Laboratory of Earth Surface Processes and Resource Ecology, Faculty of Geographical Science, Beijing Normal University, Beijing, China. [2]School of Natural Resources, Faculty of Geographical Science, Beijing Normal University, Beijing, China. [3]Department of Civil and Environmental Engineering, Massachusetts Institute of Technology, Cambridge, MA, USA. [4]Department of Earth System Science, Stanford University, Stanford, CA, USA. [5]Woods Institute for the Environment and Precourt Institute for Energy, Stanford University, Stanford, CA, USA. ✉e-mail: enzaidu@bnu.edu.cn

are relatively lacking in many developing countries (e.g., China and India) with rapidly growing areas of young urban greenspaces[27]. Overall, it is challenging to accurately evaluate the size of SOC pool in urban greenspaces globally.

Surface soils (e.g., 0–20 cm) provide major amounts of nutrients for plant growth and consequent ecosystem services[28–30]. Surface soils are also strongly imprinted by plants and contain a large proportion of SOC in the whole soil profile[14,31]. To fill the abovementioned research gaps, we constructed a global database of SOC concentration and density of greenspaces within urban built-up areas (SOC-U) to a depth of 20 cm by compiling data of 420 observations from 257 cities in 52 countries (Fig. 1a; Supplementary Table 1 and 2). We first conducted linear model selection analysis and random forest analysis to evaluate the relative importance of climatic, vegetational, social-economical, and topographical factors in explaining the spatial variation of the surface SOC density. We further mapped SOC density for global mid- and large cities (i.e., those with urban populations larger than 500,000 people[32]) using the random forest model which was shown to perform better than linear models (see Supplementary Table 3). Finally, we combined data on the area of urban greenspaces (reference year 2015) and SOC density to estimate the total SOC stocks in global urban greenspaces. We also compared our results with the estimates of SOC stocks for major natural terrestrial biomes.

## Results
### Spatial variations of SOC concentration and density
Observed values of surface SOC concentration in urban greenspaces varied considerably across cities (Supplementary Fig. 1a), ranging from 3.6 to 101.0 g C kg$^{-1}$ with a global geometric mean of 24.6 g C kg$^{-1}$ (median = 25.1 g C kg$^{-1}$). Surface SOC density of urban greenspaces also

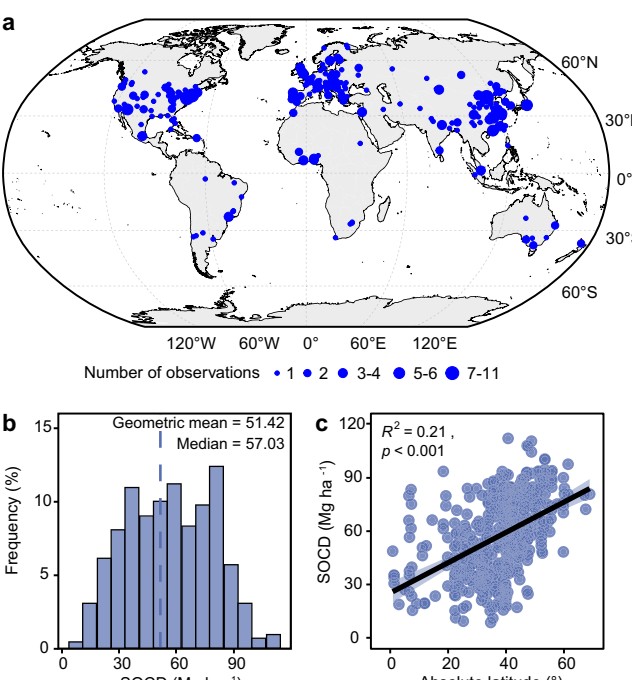

**Fig. 1 | Spatial distribution, frequency distribution and latitudinal trend of observed surface SOC density (SOCD) (0–20 cm) across global urban greenspaces in SOC-U database. a** Global distribution of observations. **b** The frequency distribution of observed SOCD. **c** Changes in observed SOCD with latitude (absolute values; See Supplementary Fig. 2a for separate analyses of northern and southern hemispheres). The size of blue circle in Fig. 1a indicates the number of reported SOC observations from different studies within each city. The dashed line in Fig. 1b indicates the geometric mean of observed SOCD. The shaded area in Fig. 1c represents the 95% confidence interval of the linear regression.

varied greatly across cities (Fig. 1b and Supplementary Table 4), ranging from 8.8 to 112.0 Mg C ha$^{-1}$ with a global geometric mean of 51.4 Mg C ha$^{-1}$ (median = 57.0 Mg C ha$^{-1}$). Both SOC concentration and density increased significantly at higher latitudes on a global scale ($p < 0.001$; Fig. 1c) and for both the northern and southern hemispheres ($p < 0.001$; Supplementary Figs. 1b and 2a). Among the climatic zones studied, mean values of surface SOC concentration and density were highest in boreal regions (SOC concentration, 53.3 ± 3.6 g C kg$^{-1}$; SOC density, 81.6 ± 2.4 Mg C ha$^{-1}$) and were three-times and twice the values in tropical regions, respectively (SOC concentration, 17.9 ± 1.5 g C kg$^{-1}$; SOC density, 42.8 ± 2.8 Mg C ha$^{-1}$) (Supplementary Figs. 1c and 2b).

### Variable importance and model selection
Both linear model selection analysis (Akaike weights) and random forest analysis (Mean Decrease Gini) were used to evaluate the relative importance of climatic (mean annual temperature, MAT; mean annual precipitation, MAP; temperature seasonality; precipitation seasonality), vegetational (urban greenness index, UGI; vegetation type, i.e., urban forest and urban lawn), social-economical (urban heat island index, UHI; GDP per capita, GDPP; population density, PD), and topographical variables (elevation). Results of both analytical approaches indicated that the spatial variation of surface SOC density was mainly explained by MAT, temperature seasonality, precipitation seasonality, and UGI (Fig. 2a, b). Specifically, conditional regression analysis showed that SOC density decreased significantly with higher MAT ($p < 0.001$; variance explained 12.5%; Fig. 2c), stronger temperature seasonality ($p < 0.001$; variance explained 2.5%; Fig. 2d) and precipitation seasonality ($p < 0.001$; variance explained 15.5%; Fig. 2e). Moreover, SOC density increased significantly with UGI ($p < 0.001$; variance explained 9.7%; Fig. 2f). Partial dependent plots indicated similar relationships between SOC density and these four important variables in comparison with the results of conditional regression analysis (Supplementary Fig. 3). Other potential predictors (i.e., MAP, vegetation type, UHI, GDPP, PD, and elevation), however, were less important in explaining the global patterns of SOC density in urban greenspaces (Fig. 2a, b).

Both linear model analysis and random forest analysis indicated that vegetation type (urban forest vs. urban lawn; available for 282 of 420 observations) had little importance for predicting the global distribution of SOC density compared with other explanatory variables (Fig. 2a, b). Considering the incomplete information on vegetation type for urban greenspaces in our dataset (Supplementary Fig. 4), we further conducted additional analyses using all 420 observations via the two approaches and confirmed the same important predictors for SOC density (Supplementary Fig. 5). In view of the fact that data on vegetation type for corresponding urban greenspaces were not available globally, we thus ignored vegetation type in following model training and global prediction of SOC density. Based on 10-fold cross-validation, the random forest model showed a better performance compared with the best linear mixed model (R²: 45.53% vs 40.19%; RMSE: 16.85 vs 17.64) (Supplementary Fig. 6 and Supplementary Table 3). We then mapped surface SOC density of global mid- and large cities using the random forest model combined with corresponding data on the nine predictors (see Methods for details).

### Global and national mapping of SOC density and stocks
Predicted SOC density of the surface soil layer (0-20 cm) generally exhibited a strong latitudinal pattern (Fig. 3a) with a global average of 55.2 Mg C ha$^{-1}$ (95% confidence interval: 51.9–58.6 Mg C ha$^{-1}$). This value is 28 % higher than the global average for all natural soils (43 Mg C ha$^{-1}$) (Table 1). Specifically, surface SOC density in urban greenspaces was on average lower than that in global boreal forests (93 Mg C ha$^{-1}$) but higher than that in tropical/subtropical forests (38 Mg C ha$^{-1}$), temperate forests (48 Mg C ha$^{-1}$), and global grasslands

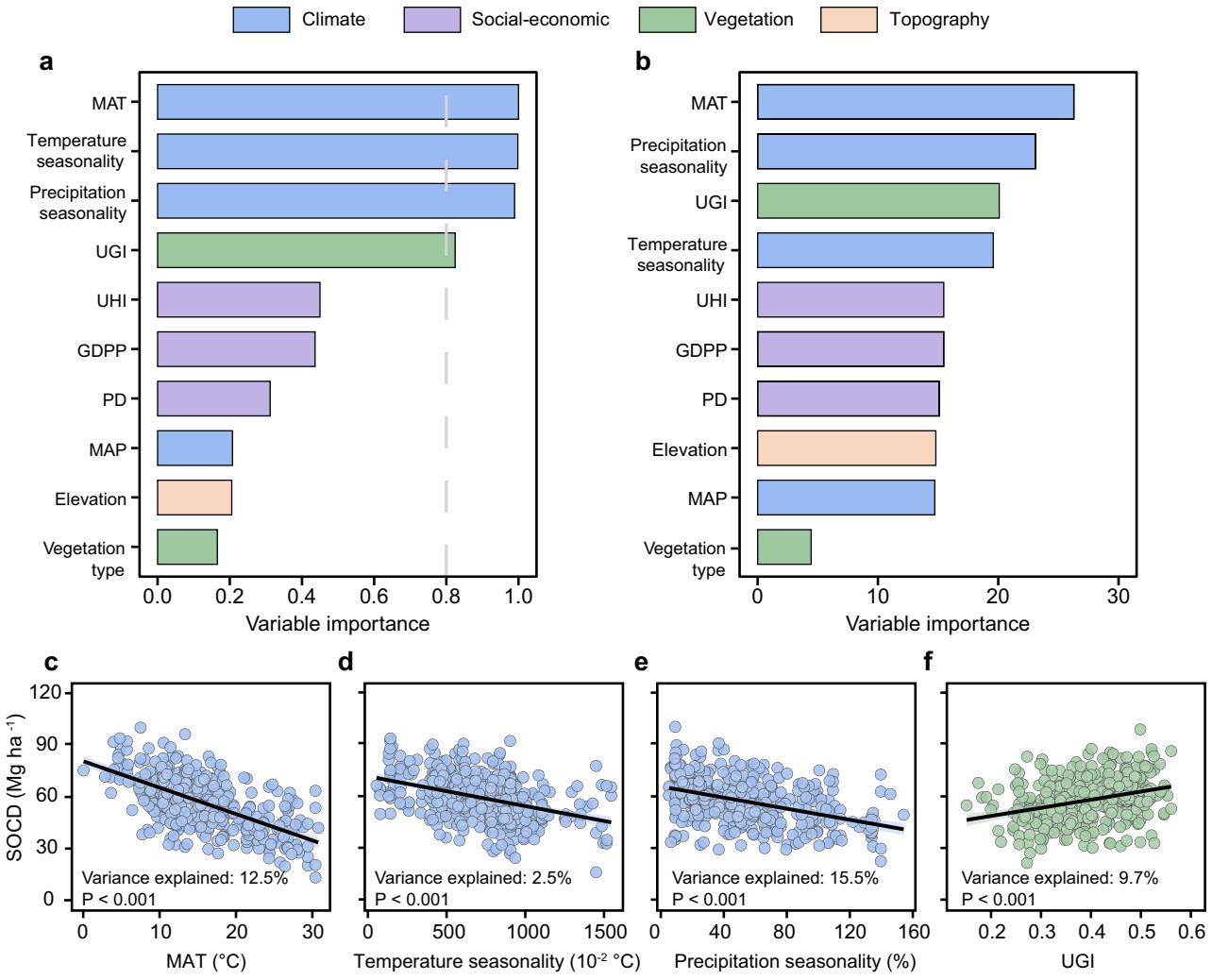

**Fig. 2 | Relative importance of the potential predictors and conditional regression plots for important predictors. a, b** Relative importance of the potential predictors for SOC density (SOCD) based on linear model analysis (**a**) and random forest analysis (**b**). **c–f** Conditional regression plots with mean annual temperature (MAT) (**c**) temperature seasonality (**d**) precipitation seasonality (**e**), and urban greenness index (UGI) (**f**). Different colours represent different predictor groups. The variable importance shown in Fig. 2a is based on the sum of the Akaike weights derived from model selection using corrected Akaike information criterion. The cut-off is set at 0.8 (grey dashed line in **a**) to differentiate among the important predictors. The importance shown in Fig. 2b is based on Mean Decrease Gini of random forest models. The black solid lines in Fig. 1c–f indicate the conditional regression fit. The shaded areas in Fig. 1c–f represent the 95% confidence intervals. Only data with reported information on vegetation type were used for the analysis ($n = 282$) and an additional analysis was also conducted using all data ($n = 420$) (Supplementary Fig. 5). MAP mean annual precipitation, GDPP GDP per capita, PD population density, UHI urban heat island index.

and shrublands (26 Mg C ha⁻¹) (Table 1). At the national scale, the top ten countries with the highest SOC density in urban greenspaces were all high-income European countries, with Ireland showing the highest value of SOC density (82.7 Mg C ha⁻¹) (Fig. 3b).

Based on the predicted SOC density and estimated areas of urban greenspaces (see Methods for details), we calculated the total SOC stocks in urban greenspaces both at city and country levels (Fig. 4a, b and Supplementary data 1, 2). The sum of surface SOC stocks in global urban greenspaces was estimated to be 1.46 Pg C (95% confidence interval: 1.37-1.54 Pg C), which was 2.7‰ of the global terrestrial SOC stocks (543 Pg C) at the same soil depth (Table 1). Notably, there were large differences in estimated SOC stocks across countries (Fig. 4b). The top ten countries with the largest SOC stocks were those having the largest areas of urban greenspaces (Fig. 4c). The United States had the largest SOC stocks in urban greenspaces (0.37 Pg C) that accounted for nearly one fourth of the global total, mainly attributable to its largest area of urban greenspaces (5.7 × 10⁴ km²; Fig. 4c). In contrast, the total SOC stocks (0.27 Pg C) in urban greenspaces in China (0.15 Pg C) and India (0.12

Pg C) only accounted for less than one fifth of the global total (Fig. 4b), although they jointly had larger areas of urban greenspaces (6.4 × 10⁴ km²) than the United States (Fig. 4c).

## Discussion

Our study provides a global assessment of the distribution and key drivers of surface SOC in urban greenspaces using newly compiled comprehensive datasets. We found that surface SOC density in urban greenspaces increased significantly at higher latitudes, consistent with the trends in natural soils[14,33]. Cities with colder climates, less climate seasonality, and higher urban greenness tend to show higher SOC density in urban greenspaces (Fig. 2c–f). The global pattern of SOC density in urban greenspaces was significantly explained by MAT, temperature seasonality and precipitation seasonality, implying a strong climatic control of large-scale variation in SOC density despite considerable human disturbances. The higher SOC density at lower MAT might be attributable to the fact that low temperature strongly limits SOC decomposition and thus favours SOC accumulation[34]. Additionally, stronger temperature seasonality and precipitation

seasonality are likely less favourable for plant growth and thus result in lower litter inputs as sources for SOC accumulation[35]. However, our analysis did not identify MAP as a statistically important predictor for the SOC density in urban greenspaces (Fig. 2a), in contrast to previous findings for natural vegetation[36]. This result is likely due to the fact that cities are generally built in relatively humid areas to provide adequate water for human demands[37] and that urban greenspaces are frequently irrigated[20], potentially eliminating water limitations to vegetation growth and SOC accumulation.

Our results provide evidence that vegetation conditions significantly influence SOC density in global urban greenspaces. We found positive effects of urban vegetation greenness (indicated by UGI) on SOC density (Fig. 2f). This finding is also supported by field studies across urban-rural gradients at local scales[38]. A higher degree of urban greeness reflects increasing coverage of urban vegetation and plant productivity, thus favouring SOC accumulation[39]. Surprisingly, we found that vegetation type and anthropogenic variables (e.g., UHI, GDPP, and PD) exerted only limited effects on the spatial pattern of SOC density globally. The low importance of vegetation type could

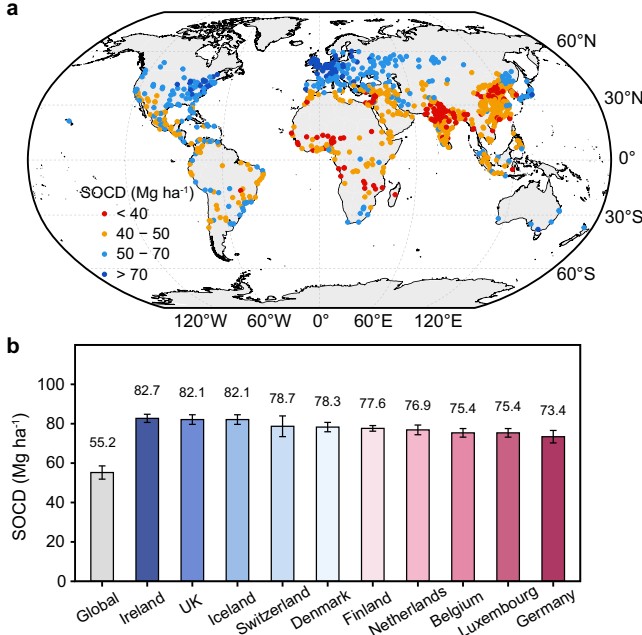

**a**

SOCD (Mg ha⁻¹)
- < 40
- 40 – 50
- 50 – 70
- > 70

**b**

**Fig. 3 | Global patterns of predicted surface SOC density (SOCD) (0–20 cm) and area-weighted national mean SOCD in urban greenspaces. a** Predicted SOCD of urban greenspaces for mid- and large cities (urban population > 0.5 million) (Supplementary data 1). **b** Average SOCD of urban greenspaces estimated for the globe and top ten countries weighted by areas. The colour of circle in Fig. 3a indicate variations in predicted SOCD. Error bars in Fig. 3b represent the 95% confidence intervals.

possibly be attributable to a coarse classification (e.g., urban forest and urban lawn) and a potentially masking effect of climate variables on the role of vegetation. The observed predominance of climatic variables over anthropogenic variables in shaping the global patterns of urban SOC density suggests that the key drivers of SOC density likely vary across spatial scales. Climatic drivers have been found to determine spatial patterns of SOC density at a continental or global scale[40,41]. In contrast, anthropogenic drivers are likely more influential to affect SOC density locally. For example, management operations in urban greenspaces (e.g., selection of plant species for urban greening, nutrient fertilization, irrigation, and pest control) can favour vegetation growth and SOC accumulation[42], but such effects may be unable to substantially alter the global pattern of SOC density.

Our data synthesis presents an assessment of surface SOC in global urban greenspaces (Figs. 3a and 4a). Although urban greenspaces only accounted for less than 3‰ of global terrestrial surface SOC stocks (0–20 cm), they are characterised by high SOC density (55.2 Mg C ha⁻¹) compared with other terrestrial biomes (Table 1), implying a large potential for SOC accumulation in urban greenspaces per unit area. Our estimate of the average SOC density in global urban greenspaces is higher than a previous result (40.2 Mg C ha⁻¹)[19], which used a small number of SOC data inputs and simply averaged the results. The relatively high SOC density observed in urban greenspaces can be explained by favourable conditions for urban plant growth and soil C accumulation[43], such as high-level atmospheric $CO_2$ concentrations, additional nutrient and water inputs from human activities[5].

We found a substantial variation in SOC stocks across countries (Fig. 4b), which was mainly related to national urban greenspace area (Fig. 4c). The United States has the largest area of urban greenspaces and corresponding SOC stocks (Fig. 4c), while China and India, holding a slightly larger area of urban greenspaces in total (6.4 × 10⁴ km² vs. 5.7 × 10⁴ km²), account for less SOC stocks than the United States (0.27 vs. 0.37 Pg C). This is likely due to the fact that a large proportion of urban greenspaces are relatively young in view of China and India's relatively rapid urbanization in recent decades[27] and thus vegetation had a limited time to establish and replenish the SOC pools. Additionally, we found large variations in SOC density across countries (Fig. 3b), generally showing highest values in wealthy countries at high latitudes. As indicated by our analysis, surface SOC density showed a significant decrease with higher MAT (Fig. 2c). Future climate warming may lead to a risk of SOC loss in urban greenspaces of countries at high latitudes, most of which hold high levels of SOC density. In contrast, many developing countries are undergoing a rapid expansion of young urban greenspaces with low SOC density due to a short time of SOC accumulation and poor management[44]. There will be a potential increase in SOC stocks in the large areas of young urban greenspaces most of which are located in developing countries. In that case, urban soils need to be properly managed to store more C and make urban ecosystems more resilient and sustainable in the context of future climate change.

**Table 1 | Estimated surface SOC density and stocks (0–20 cm) for global urban greenspaces and major terrestrial biomes**

| Biome | SOC density (Mg C ha⁻¹) (0-20 cm) | SOC stocks (Pg C) (0–20 cm) | Contribution to global SOC stocks (%) | References |
|---|---|---|---|---|
| All soils[a] | 43 | 543 | — | 14 |
| Tropical/subtropical forests[a] | 38 | 140 | 25.78 | |
| Temperate forests[a] | 48 | 144 | 26.52 | |
| Boreal forests[a] | 93 | 121 | 22.28 | |
| Grasslands and shrublands [a] | 26 | 79 | 14.55 | |
| Urban greenspaces | 55.2 (51.9–58.6) | 1.46 (1.37–1.54) | 0.27 | This study |

Numbers in parentheses indicate 95% confidence intervals in this study. [a] SOC stocks in the 0–20 cm layer is estimated by assuming that this layer contains 73.56% of the SOC stocks in the 0–30 cm layer[31]. 'All soils' include SOC stocks of all terrestrial biomes.

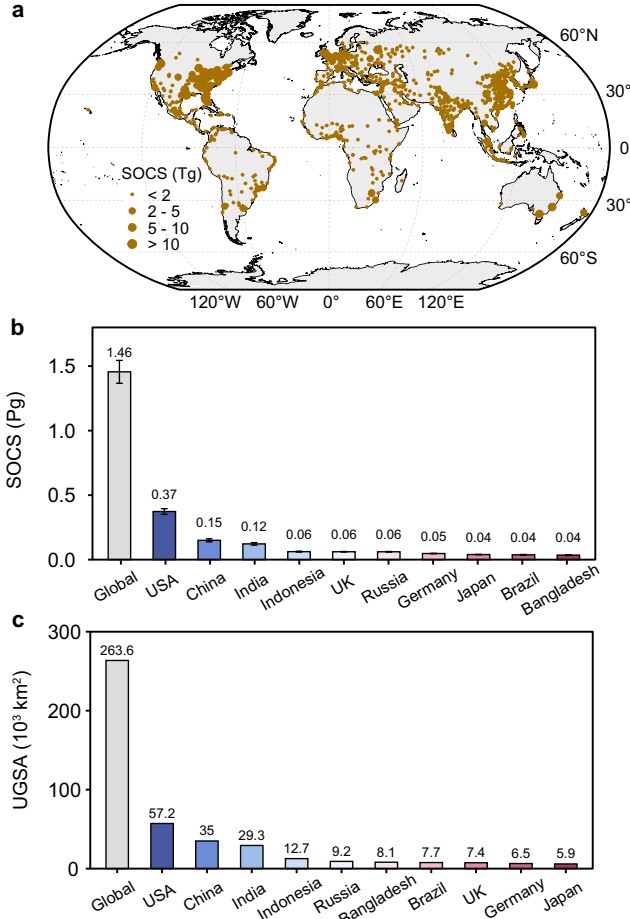

**Fig. 4 | Global patterns of surface SOC stocks (SOCS) (0-20 cm) of mid- and large cities and national estimates of SOCS in urban greenspaces. a** Predicted SOCS of urban greenspaces for mid- and large cities (urban population > 0.5 million) (Supplementary data 1). **b, c** Total SOCS (**b**) and urban greenspace areas (UGSA) (**c**) estimated for the globe and top ten countries. The size of brown circle in Fig. 4-**a** indicates the variations of predicted SOCS. Error bars in Figs. 4**b** and 4**c** represent the 95% confidence intervals. The estimates of national SOCS were based on the total national areas of urban greenspaces (Supplementary data 2).

In this study, we estimated SOC density in global urban greenspaces and quantified the roles of climatic, vegetational, social-economical, and topographical factors in explaining its spatial pattern. However, other factors may also influence SOC density, such as physicochemical soil variables, parental materials[45], and the ages of urban greenspaces. Our analyses indicated that soil pH and total nitrogen concentration significantly affected SOC concentration (Supplementary Fig. 7), but such data were relatively unavailable in global urban greenspaces, and thus, we were unable to use these inputs for our global predictions. It is also difficult to derive data for the parental materials of urban soils, many of which include transported materials from other places[46]. The ages of urban greenspaces are indicative for the length of time to accumulate SOC as topsoil organic matters are often stripped away and lost during the construction of urban greenspaces. Therefore, higher SOC density often occurs in older urban greenspaces[12,47]. Given that a large proportion of newly established urban areas and greenspaces are located in developing countries at mid- to low latitudes[27], it is possible that the ages of these urban greenspaces spatially correlate with MAT and temperature seasonality. This potential collinearity might result in an overestimation of the effects of MAT and temperature seasonality on the spatial pattern of SOC density in our analysis. Unfortunately, our ability to conduct a

quantitative assessment of the ages of urban greenspaces was constrained by the limited availability of relevant data.

High levels of nitrogen deposition and atmospheric $CO_2$ concentration in urban environments often favour plant growth and enhance SOC accumulation[48–50]. However, we were unable to conduct a quantitative analysis to incorporate such an "urban hotspot effect"[51], again due to the lack of high-resolution data within cities. Anthropogenic disturbances, such as land use change, topsoil removal, and/or import of soils from elsewhere, are common during urban expansion, but we were not able to evaluate their potential effects in our analysis. The limitation of the above-mentioned data could lead to a potential underestimation of the influence of anthropogenic factors on the spatial variations of SOC density in urban greenspaces. Nevertheless, our global prediction of SOC density in urban greenspaces generally showed low levels of uncertainty (the coefficient of variance of simulations mostly < 5%) (Supplementary Fig. 8), implying a relatively reliable estimation.

Urbanization occurs in diverse patterns (e.g., compact, dispersed, fragmented, and extensive) and this may make it challenging to accurately estimate the areas of urban greenspaces using remotely sensed data[52], potentially resulting in uncertainties of the estimates for the national and global SOC stocks in urban greenspaces. Cities contain some greenspaces where vegetation is either planted or overhang the impervious surface, while the current upscaling approach may lead to an overestimate of surface SOC stocks in these greenspaces. Moreover, it is challenging to evaluate SOC in deeper layers both due to a lack of observed data and the complexity of parental material sources. For instance, unexpectedly high SOC stocks are sometimes observed in deeper layers of urban soils due to landfill input[53]. Hence, the SOC of deeper soils in urban greenspaces needs to be studied further.

In summary, our findings elucidate the global distribution of the surface SOC in urban greenspaces as an additional share of terrestrial C budgets and provide a baseline for future urban soil C assessment. In view of the considerably high levels of surface SOC density in urban greenspaces, greening efforts likely have a potential to increase topsoil C sequestration in established urban areas. In the context of continuing urbanization and urban greening[54], total SOC stocks in global urban greenspaces will likely increase further over time in comparison to the current estimate (base year 2015). We recommend more research efforts to better understand future biogeochemical changes in global urban greenspaces and provide management options to improve the soil C storage as well as ecosystem services.

## Methods
### Data collection
By conducting a literature survey via ISI Web of Science (http://isiknowledge.com), Google Scholar (http://scholar.google.com) and China National Knowledge Infrastructure (http://www.cnki.net), we compiled a global database of surface SOC concentration and density in urban greenspaces (SOC-U) (Supplementary data 3, updated to June, 2023). The keywords 'soil organic carbon'/'SOC' and 'urban greenspaces'/'urban parks'/'urban forests'/'urban lawns' were used for literature search. Data were recorded according to the following criteria: (i) measured values of SOC concentration, SOC density or soil organic matter (SOM) concentration (further transformed to SOC concentration by multiplying SOM concentration with a factor of 0.58, which has been adopted by previous studies[55–57]) were reported; (ii) soils were sampled in urban greenspaces within the built-up areas, while data were not recorded when field sampling was conducted in rural areas; (iii) soil samples were collected within the top 20 cm depth (as reported in original literature). We only considered the surface soil layer because (i) surface soils are strongly affected by vegetation and human activities, (ii) surface soils are essential for nutrient retention and supply for plant growth, and (iii) surface SOC data are more

available in literature. Reported data for replicated samples of a same vegetation type within a same sampling site were averaged for further analysis. We also recorded geographical location (latitude and longitude), climatic condition (MAT, MAP, and climate zones), vegetation type (urban forest or urban lawn), land use type (parkland, residential area, or garden), elevation, and other soil properties (bulk density, clay fraction, pH, total nitrogen concentration, and total phosphorus concentration) when available from the original literature. Missing data for city geographical locations were further derived using Google Earth (https://earth.google.com). Overall, the SOC-U database includes 420 and 155 observations of surface SOC concentration and density from 244 independent publications, across 257 cities in 52 countries (Fig. 1a and Supplementary Table 5; also see a source reference list in Supplementary Information). The SOC-U database has been deposited in figshare at: https://doi.org/10.6084/m9.figshare.24946002.

To explore the potential predictors for the global pattern of SOC in urban greenspaces, we further retrieved data on climatic, vegetational, social-economical, and topographical factors (see a work flow for data collection and analyses in Supplementary Fig. 9). Data on temperature seasonality and precipitation seasonality were derived from WorldClim database at $1 \, km \times 1 \, km$ spatial resolution[58]. Information on urban greenness index (UGI), calculated as the average of annual highest values of NDVI for all pixels within the built-up areas[59], was derived from Global Human Settlement Layer (GHSL) Data Package[60]. We retrieved data on urban heat island index (UHI) from Global Urban Heat Island Data Set[61]. Data on urban population density (PD) and GDP per capita (GDPP) were also derived from Gridded Population of the World Version 4 (GPWv4)[62] and GHSL Data Package[60]. More details of the potential predictors used for model selection can be found in the Supplementary Information (Supplementary Table 1 and Supplementary Figs. 4 and 10–12). In addition, we derived data on potential predictors for major cities to predict global SOC density in urban greenspaces. Data on the areas of urban greenspaces and urban built-up areas were derived from GHSL Data Package[60] and used for upscaling (reference year 2015; Supplementary Table 2). Urban built-up areas were defined as regions dominated by continuous artificial impervious areas ( > 50%) or having a population density larger than 1,500 habitants per $km^2$ regardless of political boundaries[60]. Total urban built-up areas of all cities in each country were estimated based on GHSL Data Package[60] and further used to caculate total SOC stocks in each country.

## Data processing
When data on SOC concentration were reported in literature, SOC density(SOCD, Mg C ha$^{-1}$) was calculated according to Eq. (1)[36],

$$SOCD = SOCC \times BD \times T/10 \tag{1}$$

where SOCC is soil organic carbon concentration (g C kg$^{-1}$), BD is soil bulk density (g cm$^{-3}$), and T is soil thickness (cm). We established an empirical linear model between reported SOC concentration and soil bulk density (Supplementary Fig. 13a). Missing values of soil bulk density were then estimated using the empirical model (Supplementary Fig. 13b). We conducted an additional analysis only using data with reported values of soil bulk density ($n = 155$) and found similar results as that using all observations ($n = 420$) (compare Supplementary Fig. 14 and Fig. 2), implying that our approach to estimate soil bulk density was reasonable. The calculated SOC densities were further combined with literature reported data for further analysis.

To ensure the comparability of data derived from different studies, the original SOC density was standardized to 20 cm depth according to Eqs. (2) and (3)[31],

$$Y = 1 - \beta^d \tag{2}$$

$$X_{20} = \frac{1 - \beta^{20}}{1 - \beta^{d0}} \times X d_0 \tag{3}$$

where $Y$ is the cumulative proportion of the SOC density from the soil surface to depth $d$ (cm), $\beta$ is the relative rate of decrease in the SOC density with soil depth, $X_{20}$ is the SOC density in the upper 20 cm, $d_0$ is the original soil depth in each study (cm), and $X_{d0}$ is the original SOC density. We adopted the value of $\beta$ as 0.9786 which were commonly used in previous studies[63], including those for urban ecosystems[19].

## Data analysis
Linear regression analysis was used to test the spatial trends of SOC concentration and density with latitude. To identify the important predictors for SOC density, we selected four climatic variables (mean annual temperature, MAT; mean annual precipitation, MAP; temperature seasonality; precipitation seasonality), two vegetational variables (urban greenness index, UGI; vegetation type, i.e., urban forest and urban lawn), three social-economical variables (urban heat island effect, UHI; GDP per capita, GDPP; population density, PD), and one topographical variable (elevation). We used two different statistical approaches (linear mixed model and random forest model) to evaluate the relative importance of these potential predictors and used the best model to conduct a global mapping of SOC density.

We first conducted a model selection analysis using linear mixed-effects models. In view of the measurements in different years or measurements by different researchers across cities (Supplementary Table 4), we used linear mixed-effects models by the 'lme4' package[64] and tested possible random effects by the 'lmerTest' package[65]. Model selection analysis was implemented based on the corrected Akaike information criterion using 'glmulti' package[66]. The relative importance of each predictor was estimated as the sum of the Akaike weights for the linear mixed-effects models in which the predictor appeared, and a cut-off value of 0.8 was used to differentiate between the important and unimportant predictors[67]. Conditional regression plots were created using the 'visreg' package to visualize the role of each important predictor on SOC density while holding all the other important variables constant (by default the median for numeric variables)[68].

We also conducted an analysis using random forest models in combination with all ten potential predictors. We used a 10-fold cross-validated method to train the random forest models and obtained the optimal parameters (ntree, the number of decision trees; mtry, the number of input variables at each split) with the 'caret' package[69]. Using the optimal parameters, we then tuned the random forest model. The variable importance was further evaluated by Mean Decrease Gini and partial dependence plots were created using the 'pdp' package[70]. The determination coefficient ($R^2$) and root mean square error (RMSE) were used to compare the performance of linear models and random forest models. The random forest model was then used to map SOC density for global urban greenspaces in view of its better performance over linear model (higher $R^2$ and lower RMSE; Supplementary Fig. 6 and Supplementary Table 3).

To map and estimate the SOC densities and stocks in global urban greenspaces, we combined comprehensive spatial datasets for world cities (Supplementary Table 2 and Supplementary Fig. 9). We first derived data on urban greenspaces for 1039 mid- and large cities (urban population >500,000)[71], defined based on UN population statistics in 2014[32]. We also obtained data on the nine predictors from WorldClim database[58] and Global GHSL Data Package[60]. Vegetation type (urban forest vs. urban lawn) was not used in global prediction because it had a least importance in predicting the global distribution of SOC density (Fig. 2a, b) and corresponding data were not available for global urban greenspaces. The random forest model was used to predict SOC density of urban greenspaces for global mid- and large

cities (see Supplementary data 1 for a list of the cities and corresponding SOCD and SOCS). The relative uncertainty of the predicted SOC density was estimated by coefficient of variance (C.V., %) of the ten simulations when training random forest models. Note that 35 cities were excluded because of lacking data on predictors in Global GHSL Data Package.

The city-level SOC stocks ($SOCS_c$, Tg C) was calculated as Eq. (4),

$$SOCS_c = SOCD_c \times UGSA_c \times 10^{-6} \tag{4}$$

where $SOCD_c$ (Mg C ha$^{-1}$) is the predicted mean SOC density in that city and $UGSA_c$ (ha) is the corresponding area of urban greenspaces[71]. To estimate the national SOC stocks in urban greenspaces, we first estimated the national urban greenspaces area ($UGSA_n$, ha) according to Eq. (5),

$$UGSA_n = UGSA_c \times \frac{UA_n}{UA_c} \tag{5}$$

where $UGSA_c$ (ha) is the total urban greenspaces area of mid- and large cities in a country, $UA_c$ (ha) is the corresponding total urban built-up area (ha) of these cities, and $UA_n$ (ha) is the total urban built-up area of all cities in that country. The national total SOC stocks of urban greenspaces ($SOCS_n$, Tg C) was estimated as Eq. (6),

$$SOCS_n = SOCS_c \times \frac{UGSA_n}{UGSA_c} \tag{6}$$

where $SOCS_c$ (Tg C) is the sum of the SOC stocks in the mid- and large cities in that country (see Supplementary data 2 for estimated national-level SOC stocks in urban greenspaces). The global total SOC pool of urban greenspaces was estimated as the sum of the SOC stocks in all countries. The area-weighted national and global SOC density was calculated based on SOC stocks divided by its area. Data used for creating maps of the SOC density and stock in global urban greenspaces were derived from the '*rnaturalearth*' package[72]. All analyses were conducted by R 4.0.5[73].

## Data availability
All data generated in this study are available online in *figshare* at https://doi.org/10.6084/m9.figshare.24946002. A Source Data file is provided with this manuscript.

## Code availability
Data analysis was conducted using R software (Version 4.0.5) that is publicly available at https://www.rproject.org. The R codes for statistical analyses and visualization can be available from the corresponding author.

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

## Acknowledgements

This work was supported by the Fundamental Research Funds for the Central Universities, State Key Laboratory of Earth Surface Processes and Resource Ecology (2021-TS-02) and Fok Ying-Tong Education Foundation (161015).

## Author contributions

E.D. conceived the project. H.G. and E.D. compiled the database and analysed the data. H.G., E.D., R.J. and C.T. wrote and revised the manuscript.

## Competing interests

The authors declare no competing interests.
