## [Peer Review File · Nature Communications]

Global distribution of surface soil organic carbon in urban greenspacesREVIEWER COMMENTS

Reviewer #1 (Remarks to the Author):

The paper compiles SOC concentration and density data globally and explores their relation to a number of predictors, finding MAT, greenness and GDP per capita to be significant. The authors then create a statistical model to estimate urban SOC globally, providing an estimate of the global stock and its distribution across nations.

I was very excited to see data on urban soils globally being compiled and analysed as this is an overlooked SOC pool in the global budget and there is potential here to make an important contribution to our picture of global carbon cycles and to helping steer urban development and ecosystem management.

However, after consideration, I have a number of reservations and questions that lead me to recommend that the methods/supplementary need to be substantially improved to provide a rigorous replicable dataset. This is also needed to be able to interpret whether the global analysis is well-founded. A clearer discussion of limitations is also vital, and care needs to be taken with suggesting that urban development could result in SOC accumulation.

I arrange my main comments by section below.

Spatial variations of SOC concentration and density

1. The approach in theory this section seems sound and straight forward. However, I was surprised by how little data has been thrown up by the search. I know from our own studies and reading of literature nationally that there are more than 6 observations of SOC in the UK cities in the dataset and on map figure S1. Perhaps I am reading figure S1 incorrectly? It suggests that this is the number of observations but perhaps it shows the number of separate studies reporting values? In which case, how are these averaged? Some studies will have very many samples from a wide variety of spaces, and others very few or focused on a limited range of sites depending on the question being asked.
2. Generally, not enough information is given in the methods or the supplementary to be able to understand the mean values, replicate the search that produced the dataset or to trace the data. The data in the database does not seem to be referenced. Apologies if I have missed this.
3. Figure 1 would benefit from larger fonts on the map. The resolution of the font is poor making it hard to distinguish some cities. Also, I find it hard to distinguish the blue scale. For other figures, please avoid red and green combinations as they are not friendly to those with colour vision issues.
4. Apologies if I am misunderstanding, but where are the negative latitude sites on figures 1b and 1d?

5. Did you record the ecosystem/habitat/greenspace type of for each data point compiled? I imagine this has a large effect on the SOC.

Model selection and best predictors

6. GDPP: What explanation/rationale is there for GDPP being a significant predictor? Cities are not homogenous in their socioeconomic status – greenspaces across urban areas will be located across a range of socioeconomic contexts. Economic disparities will also range greatly city to city. Can you comment on what bias might there be in the data collected in the studies in sampling high economic status areas where it is more likely that the area is fertilized/irrigated/not as polluted/long established? This issue is important as it relates to how valid it is to assume that the relations derived apply across all urban greenspace within a city and the validity of the global mapping approach.

Global mapping of SOC density and stocks

7. We absolutely need global maps and stocks of urban SOC, and I appreciate the aims of the authors here and acknowledge that there will be limitations to any study that tries to do this. However, I have concerns that the approach here is a stretch of what the data can show us – this is unfortunately still a relatively small dataset of 161 points. The limitations of the study need to be more clearly and plainly stated if this type of analysis is to be included in the paper. Can the authors please comment on potential bias in the underlying data, the potential influence of habitats, the potential effects of assuming that the proportion of greenspace in medium/large cities examined is representative of all greenspaces nationally, and the extent to which it is valid to estimate this map given the paucity of data in the global south.

8. More information on what data was used from the cited spatial datasets needs to be given – line 318 onwards.

9. It is a counter-intuitive result that SOC stocks in China and India are so low, given the number of cities over the vast regions. Do you think GDPP is playing too large a role in your model to allow the model to hold across these regions?

Discussion

10. Line 184 – more nutrient additions and irrigation is not necessarily ‘better’ management of vegetation. I wonder whether what you are seeing here is the length of time an urban greenspace has been established – young cities are likely to have been recently developed, and during development topsoils and organic matter are often stripped away and lost, and vegetation has had a limited time to establish and replenish SOC pools.

11. Line 192 – good to see the estimate being contextualised within the global carbon stocks. I worry about the statement that there is a large potential for SOC accumulation given the rapid urban

expansion expected in future though. This is a potentially dangerous assertion suggesting urban development may be positive for SOC and global carbon cycles. The amount of non-greenspace being created during expansion, the degree of topsoil removal/disturbance, import of soils from elsewhere to support urban greenspaces, etc. is not being considered in this statement.

Reviewer #2 (Remarks to the Author):

This paper attempts to calculate the SOC stocks from urban areas across the globe. The authors are building on previous work on urban soils and notes the lack of global estimates. A number of different data sets and models are used to estimate and scale to the globe. In general, the writing is well structured and clear. However, I do have some major concerns with regard to the clarity and significance from this version.

Major Comments

1. I found the information in the methods and approach to be lacking, including clear definitions that are key to the study. This lack of information and clarity made it very difficult to estimate the validity and significance of the results. I think a lot of this could be addressed and explained within the methods of SI. Below the major omissions I think need to be addressed.

a. Greenspace – This term is very ambiguous and needs to be defined in a way that makes it clear what is being measured and scaled, as well as how literature SOC data supports it. The manuscript uses parks and residential gardens as examples, but that is insufficient. Greenspace could refer to parks, residential lawns, abandoned parcels, greenways, remnant forest, street trees, or any aggregation or subset thereof. I suggest a more detailed operational definition that addresses the different land covers in the city and is true to the literature data supporting it.

b. Greenspace estimates by Huang et al was used as a scaling step, which seems to be developed from vegetation coverage. Key information about that step needs to be included in the methods. Another consideration that is not addressed is that vegetation and soil extents do not match. Cities contain significant “green space” where vegetation is either planted (street trees) or overhangs impervious surface. Again, depending on the operational definition of greenspace used in this study, this might be a significant area or not. To my knowledge, we know even less about C storage under impervious surfaces.

c. The literature sources for each of the cities to be included in the SI and data. The reference list included is not clear enough. The land uses/covers sampled in that literature needs to be identified. If most of the data is coming from parks and remnant forests, but being extrapolated to residential areas and lawns, that would be a significant mismatch. Residential spaces vary greatly and literature data may not be representative. Additional useful information like the number of sites, total area, and sample density (samples per area) within each city would help provide insight into uncertainty and scope.

d. Lastly, a conceptual framework or illustration that conveys the data sources and scaling approach would be helpful. Cities are notorious for substantial heterogeneity across multiple scales. I am not convinced, perhaps due to some of the ambiguity, that the approach used yields valid results. In part, for some of the reasons in the discussion. Age is a really important factor in urban greenspace and SOC

accumulation that is not included here. It seems even the authors are suggesting GDP might be a correlated and account for some of its importance.

2. Given the lack of clarity, I am uncertain of the validity of the global estimates produced and thus the contribution this study makes to our understanding of SOC in urban landscapes. To be clear, I think the goal of generating global scale estimates of urban SOC concentrations and stock is an interesting contribution. But the finding that urban SOC still is controlled by MAT and overall greenness at the global scale is not particularly novel, nor is the estimate very different from previous. The other feedback I will provide the authors regards to significance to the relevant fields. I think there are some interesting findings for the urban soils and biogeochemistry discipline. But I am not sure I see a substantial impact on the global carbon cycling community. There is so much uncertainty in soil carbon and fluxes, constraining the small contribution of urban landscapes helps very little. The manuscript could provide a stronger argument for why this would be of interest outside of the discipline.

Minor Comments

1. Line 276 – The use of SOC to estimate Bulk density is interesting, but seems like it would introduce a lot of error. This is somewhat concerning though because SOC concentration is multiplied by BD to estimate stocks and seems like there would be a potential propagation of error. Is there enough data with measured density to determine how much difference in SOC stock may be generated using estimates vs actual data?
2. Equation 5 – Unclear where the data is from and what the definitions are for total urban area. I assume the U_a is coming from Florczyk et al., 2019, but it is not clear. Also need to define “urban” is this any settlement? Is there a lower population threshold? Not clear enough to judge.
3. SI Many of the figures in the SI are too small to read. Please make maps that are large enough to see the color gradients and circles without having to zoom in 200%.

Reviewer #3 (Remarks to the Author):

Overview and Key Results

The authors present an important first estimate of global carbon stores in the top 20 cm of soils in urban greenspaces. They note that SOC in these soils remains understudied and that urban soils are generally excluded from global analyses of soil carbon stores. This paper provides a remedy, offering an estimate of these carbon stores that confirms that urban soil management has a significant role to play in global carbon budgets. The authors also identify per capita GDP and higher greenness indices as associated with higher SOC.

Validity

Overall, the methods are appropriate and thorough. However, because urban SOC data is still relatively scarce, the authors could provide better estimates of uncertainty. In addition, a few conclusions

(specifically those regarding the influence of higher per capita GDP and higher greenness) are made that are not fully supported by the data, in part because of unexplored uncertainty (see comments below and in Suggested Improvements for more specifics).

Data on urban soil C stores are inconsistent and incomplete. The authors use a number of steps (for example, estimating soil bulk density or converting SOM to SOC or normalizing to 20 cm depth) to fill in some of these data gaps. While the methods used are appropriate, each estimate introduces more potential for error, which is then propagated through the overall estimate. In addition, while the authors note some additional factors (soil pH, N deposition, landfills, etc.) that may introduce uncertainty into these estimates, they ignore a number of other sources of error. For example, SOC data have not been collected uniformly across the globe. The U.S. and China, have more data points. Fig. 1 (b and d) depict regressions with quite low R² values that also appear to be heavily influenced by just a few data points. It would be relevant to know where these data were collected (it appears to be in equatorial Africa). Since these regressions feature prominently in the models used to estimate SOC.

Also, the extent of urban land has always been difficult to quantify. First, it changes rapidly. Second, there is not always agreement about what constitutes urban land. Third, metropolitan areas define their political boundaries in different ways (some cities include only the urban core, others annex rural land as it is developing). This also makes population counts in cities a rather arbitrary way to categorize them. Can the authors clarify this part of the methodology a bit more? Finally, patterns of urbanization take many different forms (see Schneider, A., & Woodcock, C. E. (2008). Compact, dispersed, fragmented, extensive? A comparison of urban growth in twenty-five global cities using remotely sensed data, pattern metrics and census information. *Urban Studies*, 45(3), 659-692.) The authors don't really address these challenges and do not make any estimates of how this might influence the uncertainty of results and in which direction. This topic merits considerably more discussion in the Uncertainties section.

Finally, I find it unconvincing that the global soil C baseline estimate, or the models that were used to derive it, provide any real useful guidance for soil management in cities, other than to underline that soils should be managed in the first place and not simply scraped away (a practice that releases large amounts of carbon). I think the authors could clarify this (line 32).

Significance

These are significant and important results that are essential for incorporating urban areas (where the majority of humans live) into various earth system models. This was a challenging project and will make a significant contribution to our understanding of the role of urban soils.

Data and methodology

While the supplemental files present the data used in this study, there doesn't seem to be any indication of the source of these data. I know numerous authors have compiled data from studies across the globe, but according to line 236-239, the authors did a literature search and compiled this data set. Perhaps I just missed it, but shouldn't the data set include the name of the study whence the data came? Without a source, there is no way to verify these data.

Analytical approach

The analytical approach is sound, however, there are a few instances (already mentioned) where conclusions do not seem fully supported and where uncertainty is not fully described.

Suggested improvements

Many of the suggestions below relate to topics described elsewhere in the review. They are presented in

order of appearance in the manuscript, not in order of importance.

Title and elsewhere: The authors use “distributions” in some cases in “distribution” in others. In this context, “distribution” makes more sense.

L 21-23: This one of numerous instances where the comparison context is not clear. Is this total SOC? Or SOC per greenspace land area? Or SOC per city land area?

L 24: SOC density is usually expected to include the top 1 m of soil. These are only for 0-20 cm (from Table 1, presumably) but it needs to be clarified here or it will be misinterpreted.

L 27-30: The inclusion of “the top ten countries were all high income...” immediately before listing the U.S. SOC stocks gives the impression that this high SOC stock in the U.S. is associated with the large share of the world economy represented by the U.S. and conflates the density question with the total. This should be rephrased by perhaps saying “the ten countries with the highest SOC density were all high-income countries”). I would think it most likely that the U.S. has high level of SOC in urban greenspaces because it has low density, sprawling cities with low populations and lots of greenspace.

L 32: change “management” to “assessment”. This study doesn’t tell us much about management.

L 35-37: Urban land has been quantified as being up to 3% of global land. Can the authors qualify this a bit or give a range?

L 38: “...and are generally...source” could be deleted? Seems redundant with previous sentence.

L 41: largest C pool in comparison to what? This is unclear.

L 54-59: The authors attempt seem to suggest that the urban convergence theory has been disproven by studies showing strong climate control of SOC. These aren’t really contradictory as urban convergence is more about social norms driving design and management and the outsize effects this sometimes has on plant communities, N, and water, thus causing cities to converge (not be the same, but move towards one another). Rephrase? Perhaps “However, especially at a broader scale, there is still strong...”

L 78-79: Delete? Seems like this point has already been made?

L 85 “urban built-up areas” isn’t really a very helpful definition. Can the authors elaborate?

L 90-94: the population cut-off is quite arbitrary. 500,000 is a small city in some regions and a large city in others. Can the authors justify this choice and explain the biases it might introduce? Can they indicate how the boundaries of these cities were determined (central city, metro area)?

L 97-106: This is a context issue that occurs in several places. Are these figures for density for cities? (as stated on line 100) or for urban greenspaces within cities?

L 110: I was surprised at the inclusion of soil depth here. Soil depth in cities is often related to construction practices and prior land use. It is notoriously erratic. Individual studies tend to define depth in relation to the questions of interest for that study and are thus inconsistent. Where did depth data come from and how was it defined?

L121: The authors make much of the role of GDPP on SOC density here and elsewhere. I do not find this conclusion convincing. First GDPP only explains 8.7% of the variance. Second, I am not convinced it is not correlated with latitude. In L 128 the authors say there is low multicollinearity of variables, but it is difficult for me to see how latitude and GDPP can be separated based on the cities represented in the data. Many of the top ten cities are presumably located on what was previously boreal forest, the forest cover with the highest SOC density in the world. Why did the authors not compare SOC density with the SOC density of surrounding nonurban land?

L 184-5: I’m not really sure the “better” vs “poor” management are appropriate terms. More irrigation or

N additions may result in greater SOC density, but is it really “better” management? I’m also not sure a broad generalization about developing countries can be made from the Zhang et al. 2022 paper. In addition, papers regarding SOC accumulation over time are looking at small time scales (say 30 years) and are not really applicable to comparisons between ancient urban cores and newer development where the time scale could be 100s or 1000s of years. I think the authors need to rethink some of the generalizations here.

L 211-227: Additional uncertainties and their impacts should be included here. These include potential mismatches between the data sets used for the SOC vs the cities studied, questions of landuse history, etc. The authors rightly point to high levels of HAHT soils in place in cities. This trend is in fact increasing in many cities and the potential for C sequestration in some of these soils is unknown and likely quite different from traditional soils. In addition, many soils with high organic matter content are buried (buried A horizons) in some cities. Can the authors provide some estimates on the uncertainty? What was the range of SOC densities reported from the actual data vs predicted results from the models. Perhaps I missed it, but I did not see this kind of analysis in this study.

Fig 3 b and d: b shows SOC density, while d shows total stocks. I think it would be informative to have some indication of density by country or also perhaps proportion of urban land represented by a given country. Just seeing how much total is in each country isn’t as informative.

Clarity and context

In general, the paper is clearly presented. However, there are numerous locations where the scale or context is not clear, especially when comparisons are being made. While it can likely be puzzled out, a few additional words would be helpful to readers. These are noted in the Suggested improvements section.

References

References seem appropriate, except as noted elsewhere.

Reply to REVIEWERS' COMMENTS

Reply to Reviewer #1:

The paper compiles SOC concentration and density data globally and explores their relation to a number of predictors, finding MAT, greenness and GDP per capita to be significant. The authors then create a statistical model to estimate urban SOC globally, providing an estimate of the global stock and its distribution across nations. I was very excited to see data on urban soils globally being compiled and analysed as this is an overlooked SOC pool in the global budget and there is potential here to make an important contribution to our picture of global carbon cycles and to helping steer urban development and ecosystem management.

Reply: Thanks for your insightful comments. We have carefully revised our manuscript according to your suggestions.

However, after consideration, I have a number of reservations and questions that lead me to recommend that the methods/supplementary need to be substantially improved to provide a rigorous replicable dataset. This is also needed to be able to interpret whether the global analysis is well-founded. A clearer discussion of limitations is also vital, and care needs to be taken with suggesting that urban development could result in SOC accumulation.

Reply: We have substantially revised the manuscript according to your comments.

First, we have included more details of the method to compile the database of SOC concentration and density in global urban greenspaces (SOC-U). We have further updated the database and our SOC-U database currently includes 420 observations from 257 cities in 52 countries. The sample size has been substantially increased in comparison to the earlier version of the database (n=420 vs. 161). Now the whole database and corresponding references have been provided as supplementary data and will be formally archived in *Figshare* (<https://figshare.com/s/d03ab1150d644f2df301>) when our manuscript is accepted for publication. We also conducted a reanalysis based on the updated database and revised the manuscript accordingly. Please see more details

in the revised manuscript and our responses to your specific comments below.

Second, we have now thoroughly discussed limitations of our global prediction of SOC density in the Section “Uncertainties and Implications”, such as i) not including physicochemical soil variables, parental materials, age of urban greenspaces, urban N deposition, CO₂ enrichment, and anthropogenic disturbances in current analysis due to a lack of corresponding data, and ii) not assessing SOC in deeper layers mainly due to a lack of observed data and the complexity of different parental material sources. Please see more details in the revised manuscript and our responses to your specific comments below.

Third, now we highlight that to our knowledge “Our findings present the first systematic assessment of surface-layer SOC in global urban greenspaces and provide a baseline for future urban soil C assessment in the context of continuing urbanization”. We didn’t intend to indicate that urban development could result in more SOC accumulation. We have now mentioned that “Greening efforts have a potential to increase topsoil C sequestration in established urban areas”. Sorry for any confusion in our earlier version of manuscript. We believe that we have now well addressed all your concerns. See more details below.

I arrange my main comments by section below.

Spatial variations of SOC concentration and density

1. The approach in theory this section seems sound and straight forward. However, I was surprised by how little data has been thrown up by the search. I know from our own studies and reading of literature nationally that there are more than 6 observations of SOC in the UK cities in the dataset and on map figure S1. Perhaps I am reading figure S1 incorrectly? It suggests that this is the number of observations but perhaps it shows the number of separate studies reporting values? In which case, how are these averaged? Some studies will have very many samples from a wide variety of spaces, and others very few or focused on a limited range of sites depending on the question being asked.

Reply: Thanks for your helpful suggestion. In the previous version of the manuscript,

we mainly focused on soil SOC in urban forest as it is the most important component of urban greenspace. That's why our previous version of database only included limited observations in UK. However, we have now substantially updated the database by using more combinations of key words (i.e., 'soil organic carbon' / 'SOC' and 'urban greenspaces' / 'urban parks' / 'urban forests' / 'urban lawns'). We have also recorded the land use type (i.e., parkland, residential area, and garden) / vegetation type (i.e., urban forest and urban lawn) of each greenspace from the original literature (Fig. 1; also see below). Finally, our updated database includes 420 observations from 257 cities in 52 countries (Fig. 1a). Specifically, our database includes 21 observations from 14 cities in the U.K. Now the whole database and corresponding references have been provided as supplementary data and will be formally archived in *Figshare* (<https://figshare.com/s/d03ab1150d644f2df301>) when our manuscript is accepted for publication. We are sorry for any confusion about the observations in Figure S1 of the previous version. We have now updated it and included it as Fig. 1a in the main text (also see it below). The size of the blue circle indicates the number of SOC observations from different locations within a city. As you can see from the updated figure, the current database includes data from major cities globally.

We have also clarified in the Methods section that we recorded data according to the following criteria: (i) measured values of SOC concentration, SOC density or SOM concentration (further transformed to SOC concentration) were reported; (ii) soils were sampled in urban greenspaces within the built-up areas, while data were not recorded when field sampling was conducted in the rural areas within city boundary; (iii) soil samples were collected from the top 20 cm depth. Reported data for replicated samples of a same vegetation type within a sampling site were averaged for further analysis. We have now described the additional details in the method section.

Fig. 1 Spatial distribution, frequency distribution and latitudinal trend of observed surface (0-20 cm) SOC density (SOC-D, Mg C ha⁻¹) across global urban greenspaces in SOC-U database. a, Global distribution of observed SOC-D. **b,** The frequency distribution of observed SOC-D. **c,** Changes in observed SOC-D with latitude (absolute values; See Supplementary Fig. S2a for separate analyses of northern and southern hemispheres). The size of blue circle in Fig. 1a indicates the number of reported SOC values from different studies within each city. Shaded areas in Fig. 1c represent 95% confidence intervals.

2. Generally, not enough information is given in the methods or the supplementary to be able to understand the mean values, replicate the search that produced the dataset or to trace the data. The data in the database does not seem to be referenced. Apologies if I have missed this.

Reply: Thanks for your helpful comments. In the revised method section, we have now

described the database in more detail to improve the reproducibility of the current study (also see the text below). Additionally, references have now been provided and linked to the observations in the database (See more details in Supplement information and database). Now the whole database and corresponding references have been provided as supplementary data and will be formally archived in *Figshare* (<https://figshare.com/s/d03ab1150d644f2df301>) for further reuse of data in future studies.

3. Figure 1 would benefit from larger fonts on the map. The resolution of the font is poor making it hard to distinguish some cities. Also, I find it hard to distinguish the blue scale. For other figures, please avoid red and green combinations as they are not friendly to those with colour vision issues.

Reply: Thanks for your suggestions. In the revised manuscript, we have now updated all figures/maps by using larger fonts and avoiding combinations of red and green colors. We have now used the size of blue circle as the scale for Figure 1a (also see above). See more details in our updated figures in the main text and supplement.

4. Apologies if I am misunderstanding, but where are the negative latitude sites on figures 1b and 1d?

Reply: Sorry for any confusion. We used absolute values of latitude in the regression analysis in Figure 1 because SOC density in the north- and south hemisphere showed similar latitudinal trends. We have now included an analysis for the north- and south hemisphere in the revised supplementary information (Supplementary Fig. 2, also see below). The results show that SOCD in the north- and south hemispheres both increased significantly towards higher latitudes. We have now revised the manuscript accordingly.

Supplementary Fig. 2. Changes in surface SOC density (SOC) of urban greenspaces towards higher latitudes in the northern hemisphere (a) and southern hemisphere (b).

5. Did you record the ecosystem/habitat/greenspace type of for each data point compiled? I imagine this has a large effect on the SOC.

Reply: Thanks for your helpful suggestion. We didn't include the type of vegetation in our previous analysis. We have now updated our database and retrieved information on the type of greenspace (mainly urban forest and urban lawn; park, residential area and garden; Supplementary Fig. 4; also see below). Using the updated database, we have reanalyzed the importance of the explanatory variables (including the vegetation type and other climatic, vegetation, anthropogenic and topographical variables). The results based on linear model analysis and random forest analysis both show that vegetation type is the least important predictor of SOC in urban greenspaces on a global scale that we tested (Fig. 2a & 2b; also see below). We have now updated the manuscript accordingly.

Supplementary Fig. 4. Sample numbers for different vegetation types of the SOC-U database.

Fig. 2. Estimated importance of explanatory variables based on model selection analysis (sum of the Akaike weights for the linear mixed-effects models in which the explanatory variable appeared) (a) and random forest model analysis (Mean Decrease Gini of random forest models) (b). See the revised method section for more details.

Model selection and best predictors

6. GDPP: What explanation/rationale is there for GDPP being a significant predictor? Cities are not homogenous in their socioeconomic status – greenspaces across urban areas will be located across a range of socioeconomic contexts. Economic disparities will also range greatly city to city. Can you comment on what bias might there be in the data collected in the studies in sampling high economic status areas where it is more likely that the area is fertilized/irrigated/not as polluted/long established? This issue is important as it relates to how valid it is to assume that the relations derived apply across all urban greenspace within a city and the validity of the global mapping approach.

Reply: In the previous version of the manuscript, we assumed that urban green spaces are generally better managed in cities with higher GDPP and better management may favor urban forest growth and soil C accumulation. We fully agree that cities are not homogenous in their socioeconomic status and economic disparities also vary greatly among cities. Using an updated database of a larger sample size (420 vs. 161), we have now conducted a reanalysis by including four climatic variables (mean annual temperature, MAT; mean annual precipitation, MAP; temperature seasonality; precipitation seasonality), two vegetational variables (urban greenness index, UGI; vegetation type, i.e., urban forest and urban lawn) three anthropogenic variables (urban heat island effect, UHI; GDP per capita, GDPP; population density), and one topographical variable (elevation). The new results based on two different approaches both show that GDPP played less important role in explaining the global patterns of SOC density (Fig. 2; also see above). The linear mixed model analysis clearly indicates that GDPP didn't significantly improve the model performance (Table. R1, also see below). This implies that our previous results might be biased. We have now revised our manuscript accordingly. We thank the reviewer again for pointing out this important issue.

Models:

Mod 1: $\text{SOCD20} \sim \text{MAT} + \text{T_seasonality} + \text{P_seasonality} + \text{UGI} + (1 \mid \text{city})$

Mod 2: $\text{SOCD20} \sim \text{MAT} + \text{T_seasonality} + \text{P_seasonality} + \text{UGI} + \text{GDPP} + (1 \mid \text{city})$

Table. R1. A summary of model comparison showing no significant of model performance by including GDPP as a predictor.

	npar	AIC	BIC	logLik	deviance	Chisq	Df	Pr(>Chisq)
Mod 1	7	3578.9	3607.1	-1782.4	3564.9			
Mod 2	8	3579.8	3612.2	-1781.9	3563.8	1.0071	1	0.3156

Global mapping of SOC density and stocks

7. We absolutely need global maps and stocks of urban SOC, and I appreciate the aims of the authors here and acknowledge that there will be limitations to any study that tries to do this. However, I have concerns that the approach here is a stretch of what the data can show us – this is unfortunately still a relatively small dataset of 161 points. The limitations of the study need to be more clearly and plainly stated if this type of analysis is to be included in the paper. Can the authors please comment on potential bias in the underlying data, the potential influence of habitats, the potential effects of assuming that the proportion of greenspace in medium/large cities examined is representative of all greenspaces nationally, and the extent to which it is valid to estimate this map given the paucity of data in the global south.

Reply: We have substantially improved our study according to your comments. First, we have included a more detailed description of the dataset and statistical analysis in the revised method section. We have updated the database (including 420 observations from 257 cities in 52 countries; to June, 2023) and further conducted a reanalysis. The database now included a larger sample size in comparison to its earlier version (420 vs. 161). We believe that our current analysis has substantially improved.

Second, we now thoroughly discuss potential limitations in the Section Uncertainties and implications. Using the updated database, we conducted a quantitative analysis of the impact of the reviewer’s hypotheses about the role of habitat type. Our reanalysis shows that vegetation type is not an important predictor of the global pattern of SOC density (Fig. 2). The result implies our global prediction of SOC

density is reasonable although habitat/vegetation types were not included as a predictor due to a lack of corresponding data. We have now also discussed other potential uncertainties due to assuming that the proportion of greenspace in medium/large cities examined is representative of all greenspaces nationally. We have collected data as comprehensively as possible, but there is still relatively little data from the southern hemisphere (32/420). This finding is common across meta-analyses and also highlights the need for more research in the global south.

8. More information on what data was used from the cited spatial datasets needs to be given – line 318 onwards.

Reply: We have now included more detailed descriptions of data used for model analysis and global mapping of SOC density. Please see more details in the main text.

9. It is a counter-intuitive result that SOC stocks in China and India are so low, given the number of cities over the vast regions. Do you think GDPP is playing too large a role in your model to allow the model to hold across these regions?

Reply: We have now updated the SOC-U database by including more sampled sites and conducted a reanalysis by including more predictors such as vegetation type, temperature seasonality, and precipitation seasonality. The new results show that GDPP is not an important predictor for SOC density (Fig. 2; also see above). Our reanalysis also shows that random forest model performs much better than linear models using a larger sample size (Supplementary Fig. 6; also see below). Therefore, we predicted global SOC density in greenspaces using random forest model trained by climate, socio-economic, topography and vegetation variables.

The new results further show that urban greenspace SOC density is relatively low in China and India (Fig. 3, also see below). This is likely due to the fact that China and India are located in relatively warmer climate zones and have large areas of urban greenspaces with relatively young ages due to the rapid urbanization in recent years. When multiplying by the urban greenspace area, urban SOC stocks in China and India ranked second and third, respectively, after the United States (Fig. 4, also see below).

We have revised the main text accordingly.

Supplementary Fig. 6. A comparison of performances between linear model and random forest model.

Fig. 3 Global patterns of predicted surface SOC density (SOCD) (0-20 cm) and area-weighted national mean SOC density (SOCD) in urban greenspaces. **a**, Predicted SOCD of urban greenspaces for mid- and large cities (urban population > 0.5 million) (Supplementary data 1). **b**, Average SOCD of urban greenspaces estimated for the globe and top-10 countries weighted by national areas. Error bars indicate the 95th

and 5th percentiles.

Fig. 4 Global patterns of surface SOC stocks (SOCS) (0-20 cm) of mid- and large cities and national estimates of SOCS in urban greenspaces. **a**, Predicted SOCS of urban greenspaces for mid- and large cities (urban population > 0.5 million) (Supplementary data 1). **b-c**, Total SOCS (**b**) and urban greenspace areas (UGSA) (**c**) estimated for the globe and top-10 countries. The estimates of national SOCS were based on the total national areas of urban greenspaces (Supplementary data 2). Error bars indicate the 95th and 5th percentiles.

Discussion

10. Line 184 – more nutrient additions and irrigation is not necessarily ‘better’ management of vegetation. I wonder whether what you are seeing here is the length of time an urban greenspace has been established – young cities are likely to have been recently developed, and during development topsoils and organic matter are often stripped away and lost, and vegetation has had a limited time to establish and replenish SOC pools.

Reply: We have now removed the statements on management. We also agree with your perspective that the age of the urban greenspaces is likely to be important for SOC accumulation. We now discuss this as an important potential cause for the relatively lower SOC observed in developing countries. However, we were unable to derive information on the ages of specific urban greenspaces, so we could not include it in our statistical analysis. We have now also highlighted future research efforts to assess the role of the ages of greenspaces in the revised manuscript.

11. Line 192 – good to see the estimate being contextualised within the global carbon stocks. I worry about the statement that there is a large potential for SOC accumulation given the rapid urban expansion expected in future though. This is a potentially dangerous assertion suggesting urban development may be positive for SOC and global carbon cycles. The amount of non-greenspace being created during expansion, the degree of topsoil removal/disturbance, import of soils from elsewhere to support urban greenspaces, etc. is not being considered in this statement.

Reply: Sorry for any misunderstanding. We have now revised our statements as “they are characterized by high SOC density (55 Mg C ha⁻¹) in comparison with other terrestrial biomes (Table 1), implying a considerable potential for SOC accumulation per unit area in established urban greenspaces”. Moreover, we have also discussed this in the discussion section that “Greening efforts thus have a potential to increase topsoil C sequestration in established urban areas.”

Reply to Reviewer #2:

This paper attempts to calculate the SOC stocks from urban areas across the globe. The authors are building on previous work on urban soils and notes the lack of global estimates. A number of different data sets and models are used to estimate and scale to the globe. In general, the writing is well structured and clear. However, I do have some major concerns with regard to the clarity and significance from this version.

Reply: Thanks for your helpful comments. We have thoroughly revised the manuscript according to your input. We believe that all your concerns have been well addressed. Please see more details in our reply to your comments below.

Major Comments

1. I found the information in the methods and approach to be lacking, including clear definitions that are key to the study. This lack of information and clarity made it very difficult to estimate the validity and significance of the results. I think a lot of this could be addressed and explained within the methods of SI. Below the major omissions I think need to be addressed.

Reply: We have now included more detailed information on the database of SOC concentration and density in global urban greenspaces (SOC-U) and statistical analyses in the method section. We have also clarified definitions for key terms in the main text at first appearance. We further updated the SOC-U database, conducted re-analyses and revised the manuscript accordingly. Details on the database, methodology, and statistical results have been provided in the main text or supplement.

a. Greenspace – This term is very ambiguous and needs to be defined in a way that makes it clear what is being measured and scaled, as well as how literature SOC data supports it. The manuscript uses parks and residential gardens as examples, but that is insufficient. Greenspace could refer to parks, residential lawns, abandoned parcels, greenways, remnant forest, street trees, or any aggregation or subset thereof. I suggest a more detailed operational definition that addresses the different land covers in the city and is true to the literature data supporting it.

Reply: In the revised manuscript, we have now updated our database and retrieved information on the type of greenspace, mainly including urban forests and lawns from parks, residential areas and gardens (Supplementary Fig. 4; also see below). The database includes major types of urban greenspaces that are further used for global mapping and upscaling. Our analyses based on linear model and random forest both suggest that vegetation type is not an important predictor of SOC density in urban greenspaces globally (Fig. 2a & 2b; also see below). Therefore, the global mapping of SOC density is not likely subject to influences from the type of greenspace. We have updated the manuscript accordingly.

Supplementary Fig. 4. Sample numbers for different vegetation types of the SOC-U database

Fig. 2 Relative importance of the potential predictors and conditional regression plots for important predictors. a-b, Relative importance of the potential predictors for SOC density (SOCD) based on linear model analysis (a) and random forest analysis (b). **c-f,** Conditional regression plots with MAT (c), temperature seasonality (d), precipitation seasonality (e), and UGI (f). The variable importance shown in Fig. 2a is based on the sum of the Akaike weights derived from model selection using corrected Akaike information criterion. The cut-off is set at 0.8 (dashed line in a) to differentiate among the important predictors. The importance shown in Fig. 2b is based on Mean Decrease Gini of random forest models. Only data with reported information on vegetation type were used for the analysis (n=282) and an additional analysis was also conducted using all data (n=420) (Supplementary Fig. 5). Abbreviations: UGI, urban greenness index, MAT, mean annual air temperature; MAP, mean annual precipitation; GDPP, GDP per capita; PD, population density, UHI, urban heat island index.

b. Greenspace estimates by Huang et al was used as a scaling step, which seems to be developed from vegetation coverage. Key information about that step needs to be

included in the methods. Another consideration that is not addressed is that vegetation and soil extents do not match. Cities contain significant “green space” where vegetation is either planted (street trees) or overhangs impervious surface. Again, depending on the operational definition of greenspace used in this study, this might be a significant area or not. To my knowledge, we know even less about C storage under impervious surfaces.

Reply: We fully agree with your concerns about potential uncertainties due to the estimate of areas for urban greenspaces. Our study mainly includes urban forests and lawns from parks, residential areas and gardens that are major types of urban greenspaces (see our reply above). Potential uncertainties have now been discussed as suggested above. Additionally, our study doesn’t include the analysis of urban soil SOC under impervious surfaces.

We have now mentioned this in the discussion section and highlight that C storage under urban impervious surfaces should be further considered. “Urbanization occurs in diverse patterns (e.g., compact, dispersed, fragmented, extensive) and this may make it challenging to accurately estimate the areas of urban greenspaces using remotely sensed data (Schneider & Woodcock, 2008), potentially resulting in uncertainties of estimates for the national and global SOC stocks in urban greenspaces. Cities contain some greenspaces where vegetation is either planted (street trees) or overhang impervious surface, while the current upscaling approach may lead to an overestimate of surface SOC stocks in these greenspaces. Moreover, it is challenging to evaluate SOC in deeper layers both due to a lack of observed data and the complexity of parental material sources. For instance, unexpectedly high SOC stocks were sometimes observed in deeper layers of urban soil due to landfill input (Vasenev *et al.*, 2013). Hence, the SOC of deeper soils in urban green spaces needs to be studied further.” See more detailed discussion in the Section “**Uncertainties and implications**”.

c. The literature sources for each of the cities to be included in the SI and data. The reference list included is not clear enough. The land uses/covers sampled in that literature needs to be identified. If most of the data is coming from parks and remnant

forests, but being extrapolated to residential areas and lawns, that would be a significant mismatch. Residential spaces vary greatly and literature data may not be representative. Additional useful information like the number of sites, total area, and sample density (samples per area) within each city would help provide insight into uncertainty and scope.

Reply: We have more clearly described our approach to compile the SOC-U database. Data were recorded according to the following criteria: (i) measured values of SOC concentration, SOC density or SOM concentration (further transformed to SOC concentration) were reported; (ii) soils were sampled in urban greenspaces within the built-up areas, while data were not recorded when field sampling was conducted in the rural areas within city boundary; (iii) soil samples were collected within the top 20 cm depth (as reported in original literature). We have now updated our database and retrieved information on the type of greenspace (mainly urban forest and urban lawn; park, residential area and garden; Supplementary Fig. 4; also see below). The whole database and corresponding references have been provided as supplementary data and will be formally archived in *Figshare* (<https://figshare.com/s/d03ab1150d644f2df301>) when our manuscript is accepted for publication.

Using the updated database, we have reanalyzed the importance of the explanatory variables (including the vegetation type and other climatic, vegetation, anthropogenic and topographical variables). The results based on linear model analysis and random forest analysis both suggest that vegetation type (n=282) is not an important predictor of SOC density in urban greenspaces on a global scale (Figure 2a & 2b; also see above). The analysis using all data (n=420) showed similar results of important explanatory variables (Supplementary Fig. 5; also see below). Therefore, the global mapping of SOC density is not likely subject to types of greenspaces. We have now updated the manuscript accordingly.

Supplementary Fig. 4. Sample numbers for different vegetation types of the SOC-U database

Supplementary Fig. 5. Relative importance of the potential predictors for SOC density (SOCD) based on linear model analysis (a) and random forest analysis (b). All data of SOC-U were used for the analysis (n=420). The variable importance shown in a is based on the sum of the Akaike weights derived from model selection using corrected Akaike information criterion. The cut-off is set at 0.8 (dashed line) to differentiate among the important predictors. The importance shown in b is based on Mean Decrease Gini of

random forest models. Abbreviations: UGI, urban greenness index, MAT, mean annual air temperature; MAP, mean annual precipitation; GDPP, GDP per capita; PD, population density, UHI, urban heat island index.

d. Lastly, a conceptual framework or illustration that conveys the data sources and scaling approach would be helpful. Cities are notorious for substantial heterogeneity across multiple scales. I am not convinced, perhaps due to some of the ambiguity, that the approach used yields valid results. In part, for some of the reasons in the discussion. Age is a really important factor in urban greenspace and SOC accumulation that is not included here. It seems even the authors are suggesting GDP might be a correlated and account for some of its importance.

Reply: Thanks for your suggestions. We have now included a framework that conveys the data sources and scaling approach (Supplementary Fig. 9; also see below). We have now also clearly described the approaches used for statistical analysis and upscaling. Using an updated version of SOC-U database, we conducted re-analyses and found that random forest model performs much better than linear models (Supplementary Fig. 6; also see below). Therefore, we predicted global SOCD in greenspaces using random forest model trained by climate, socio-economic, topography and vegetation variables (Fig. 3; also see below). Meanwhile, we estimated uncertainties of the prediction as the coefficient of variance (C.V., %) of predicted values using a 10-fold cross-validated method (Supplementary Fig. 10; also see below). The results confirmed robust results of the prediction.

Furthermore, we have now discussed the potentially uncertainties of our global mapping of SOC density regarding to ages of urban greenspaces. Specifically, we discussed that “The ages of urban greenspaces are indicative for the length of time to accumulate SOC as topsoil organic matters are often stripped away and lost during the construction of urban greenspaces, but we were unable to derive information on the ages of urban greenspaces for a quantitative analysis.” Moreover, our reanalysis using the updated SOC-U database indicate that GDPP played a less important role in explaining the global patterns of SOC density (see Fig. 2 & Supplementary Fig. 5; also

see above). This implies that our previous results might be biased. We have now revised our manuscript accordingly. Thanks for your inputs to address this issue.

Supplementary Fig. 9. The framework that conveys the data sources and scaling approach for national and global surface-layer SOC stocks in urban greenspaces.

Supplementary Fig. 6. A comparison of performances between linear model and random forest model.

Fig. 3 Global patterns of predicted surface SOC density (SOCD) (0-20 cm) and area-weighted national mean SOC density (SOCD) in urban greenspaces. **a**, Predicted SOCD of urban greenspaces for mid- and large cities (urban population > 0.5 million) (Supplementary data 1). **b**, Average SOCD of urban greenspaces estimated for the globe and top-10 countries weighted by national areas. Error bars indicate the 95th and 5th percentiles.

Supplementary Fig. 8. Coefficients of variance (CV, %) of predicted surface-layer (0-20 cm) soil organic carbon density (SOCD) in global urban greenspaces.

2. Given the lack of clarity, I am uncertain of the validity of the global estimates produced and thus the contribution this study makes to our understanding of SOC in urban landscapes. To be clear, I think the goal of generating global scale estimates of urban SOC concentrations and stock is an interesting contribution. But the finding that urban SOC still is controlled by MAT and overall greenness at the global scale is not particularly novel, nor is the estimate very different from previous. The other feedback I will provide the authors regards to significance to the relevant fields. I think there are some interesting findings for the urban soils and biogeochemistry discipline. But I am not sure I see a substantial impact on the global carbon cycling community. There is so much uncertainty in soil carbon and fluxes, constraining the small contribution of urban landscapes helps very little. The manuscript could provide a stronger argument for why this would be of interest outside of the discipline.

Reply: Thanks for your comments and the time invested in improving our manuscript. We have revised our manuscript according to your comments.

First, we have further emphasized the significance of our work. Our data synthesis presents to our knowledge the first systematic assessment of SOC in global urban greenspaces. Our findings elucidate the global distribution of the topsoil organic C in global greenspaces as an additional share of for terrestrial C budgets and provide a baseline for future urban soil C assessment. This topic is of growing importance in the context of continuing urbanization globally. We believe that our study is an important contribution to this field.

Second, we have now clearly described the datasets and approach used for the statistical analysis. We have also updated our SOC-U database and conducted a reanalysis with better validation and uncertainty estimates for the results. We believe that we have well addressed your concerns about the uncertainties. Please see more details in the revised manuscript and our responses to other comments.

Minor Comments

1. Line 276 – The use of SOC to estimate Bulk density is interesting, but seems like it would introduce a lot of error. This is somewhat concerning though because SOC

concentration is multiplied by BD to estimate stocks and seems like there would be a potential propagation of error. Is there enough data with measured density to determine how much difference in SOC stock may be generated using estimates vs actual data?

Reply: We understand your concerns about the estimate of bulk density when such data were missing. In our updated SOC-U data, 155 of the 420 observations have information on bulk density. We have now evaluated the empirical model and the potential uncertainties of the estimates using the updated database (Supplementary Fig. 13, also see below). We conducted an additional analysis only using data with reported values of bulk density (n=155) and found similar results as that using all observations (n=420) (compare Supplementary Fig. 14 and Fig. 2; See Supplementary Fig. 14 as below), implying that our approach to estimate bulk density was reasonable and might not introduce unacceptable errors. We have also revised the manuscript accordingly.

Supplementary Fig. 13. Relationships between observed surface-layer (0-20 cm) SOC concentration (SOCC, g kg⁻¹) and soil bulk density (g cm⁻³) in urban greenspaces (a). Predicted versus observed values of soil bulk density (b). The 1:1 line is plotted in Figure S9b.

Supplementary Fig. 14. Relative importance of the potential predictors for SOC density (SOCD) based on linear model analysis (a) and random forest analysis (b). Only data with reported information on bulk density were used for the analysis (n=155). The variable importance shown in a is based on the sum of the Akaike weights derived from model selection using corrected Akaike information criterion. The cut-off is set at 0.8 (dashed line) to differentiate among the important predictors. The importance shown in b is based on Mean Decrease Gini of random forest models. Abbreviations: UGI, urban greenness index, MAT, mean annual air temperature; MAP, mean annual precipitation; GDPP, GDP per capita; PD, population density, UHI, urban heat island index.

2. Equation 5 – Unclear where the data is from and what the definitions are for total urban area. I assume the U_a is coming from Florczyk et al., 2019, but it is not clear. Also need to define “urban” is this any settlement? Is there a lower population threshold? Not clear enough to judge.

Reply: Thanks and sorry for any confusion. We now clearly define “urban built-up areas” in the revised manuscript. “Urban built-up areas were defined as regions dominated by continuous artificial impervious areas (>50%) or having a population density larger than 1500 habitants per km² regardless of political boundaries (Florczyk et al., 2019).” Data on urban built-up areas of cities in each country were derived from Global Human Settlement Layer (GHSL) Data Package (Florczyk et al., 2019). We

have now included more detailed description in the revised method section.

References

Florczyk AJ, Corbane C, Ehrlich D, Freire S, Kemper T, Maffenini L, Melchiorri M, Pesaresi M, Politis P, Schiavina M. 2019. GHSL data package 2019. Luxembourg, EUR 29788(10.2760): 290498.

3. SI Many of the figures in the SI are too small to read. Please make maps that are large enough to see the color gradients and circles without having to zoom in 200%.

Reply: We have now checked and updated all the figures both in the main manuscript and supplementary information. Many thanks for your suggestions.

Reply to Reviewer #3:

Overview and Key Results

The authors present an important first estimate of global carbon stores in the top 20 cm of soils in urban greenspaces. They note that SOC in these soils remains understudied and that urban soils are generally excluded from global analyses of soil carbon stores. This paper provides a remedy, offering an estimate of these carbon stores that confirms that urban soil management has a significant role to play in global carbon budgets. The authors also identify per capita GDP and higher greenness indices as associated with higher SOC.

Reply: Thanks for your helpful comments. We have thoroughly revised the manuscript according to your comments. We believe that the manuscript has been substantially improved with your inputs. Please see more details below.

Validity

Overall, the methods are appropriate and thorough. However, because urban SOC data is still relatively scarce, the authors could provide better estimates of uncertainty. In addition, a few conclusions (specifically those regarding the influence of higher per capita GDP and higher greenness) are made that are not fully supported by the data, in part because of unexplored uncertainty (see comments below and in Suggested Improvements for more specifics).

Reply: We understand your concerns about the limited sample size, inappropriate effect of GDP per capita, and unexplored uncertainty. We have now well addressed these issues in the revised manuscript.

First, we have updated the SOC-U database (now including 420 observations from 257 cities in 52 countries) and included more information such as vegetation type, temperature seasonality, and precipitation seasonality. The sample size has increased substantially in our updated database in comparison to its earlier version (420 vs. 161). The updated SOC-U database thus supports more robust statistical analyses and global mapping of SOC density in urban greenspaces.

Second, we have now conducted a reanalysis using the updated database and included more explanatory variables (e.g., vegetation type, temperature seasonality, and precipitation seasonality). Different from the previous results based on smaller sample size, our reanalysis shows that GDPP is not an important predictor for global patterns of topsoil SOC density in urban greenspaces (Fig. 2a & 2b; also see below). Additionally, our results also show that vegetation type is not an important predictor of the global pattern of SOCD (Fig. 2a & 2b), implying limited influence of habitats on our global prediction of SOCD in urban greenspaces.

Third, our reanalysis also shows that random forest model performs much better than linear models (Supplementary Fig. 6; also see below). Therefore, we predicted global SOCD in greenspaces using random forest model trained by climate, socio-economic, topography and vegetation variables (Fig. 3; also see below). Meanwhile, we estimated uncertainties of the prediction as the coefficient of variance (C.V., %) of predicted values using a 10-fold cross-validated method (Supplementary Fig. 8; also see below). The results confirmed robust results of the prediction. We have also revised the manuscript accordingly.

Fig. 2. Estimated importance of explanatory variables based on model selection analysis (sum of the Akaike weights for the linear mixed-effects models in which the explanatory variable appeared) (a) and random forest model analysis (Mean Decrease Gini of random forest models) (b). See the revised method section for more details.

Supplementary Fig. 6. A comparison of performances between linear model and random forest model.

Fig. 3 Global patterns of predicted surface SOC density (SOCD) (0-20 cm) and area-weighted national mean SOC density (SOCD) in urban greenspaces. **a**, Predicted SOCD of urban greenspaces for mid- and large cities (urban population > 0.5 million) (Supplementary data 1). **b**, Average SOCD of urban greenspaces estimated for the globe and top-10 countries weighted by national areas. Error bars indicate the 95th and 5th percentiles.

Supplementary Fig. 8. Coefficients of variance (CV, %) of predicted surface-layer (0-20 cm) soil organic carbon density (SOC) in global urban greenspaces.

Data on urban soil C stores are inconsistent and incomplete. The authors use a number of steps (for example, estimating soil bulk density or converting SOM to SOC or normalizing to 20 cm depth) to fill in some of these data gaps. While the methods used are appropriate, each estimate introduces more potential for error, which is then propagated through the overall estimate. In addition, while the authors note some additional factors (soil pH, N deposition, landfills, etc.) that may introduce uncertainty into these estimates, they ignore a number of other sources of error. For example, SOC data have not been collected uniformly across the globe. The U.S. and China, have more data points. Fig. 1 (b and d) depict regressions with quite low R² values that also appear to heavily influenced by just a few data points. It would be relevant to know where these data were collected (it appears to be in equatorial Africa). Since these regressions feature prominently in the models used to estimate SOC.

Reply: Thanks for your comments. We understand your concerns about sample size, uneven distribution of data and potential uncertainties of SOC density estimates due to methodologies. In the revised manuscript, we have updated the database by a more thorough search of literature and the current sample size has increased substantially in comparison to previous version (Fig. 1a, also see below). We have also included more details on the data collection to guarantee data quality and reproducibility. Data were recorded according to the following criteria: (i) SOC concentration, SOC density or

SOM concentration (further transformed to SOC concentration) were reported; (ii) soils were sampled in urban greenspaces within the built-up areas, (iii) soil samples were collected from the top 20 cm depth. Such efforts guarantee a replication of the study and reasonable quality of our database.

Using the updated database, our reanalysis shows a more statistically robust latitudinal trend of SOC density. We understand that there are still limited data for global south. We conducted a separate analysis for the north- and south hemisphere in the revised supplementary information (Supplementary Fig. 2, also see below). The results further confirm that SOC density in the north- and south hemisphere both increased significantly towards higher latitudes. Additionally, we agree with the reviewer that there are potentially remaining uncertainties. We have now thoroughly discussed them in the revised Section “Uncertainties and implication.”

Fig. 1 Spatial distribution, frequency distribution and latitudinal trend of observed surface (0-20 cm) SOC density (SOCD, Mg C ha⁻¹) across global urban greenspaces in SOC-U database. **a**, Global distribution of observed SOCD. **b**, The frequency distribution of observed SOCD. **c**, Changes in observed SOCD with latitude

(absolute values; See Supplementary Fig. S2a for separate analyses of northern and southern hemispheres). The size of blue circle in Fig. 1a indicates the number of reported SOC values from different studies within each city. Shaded areas in Fig. 1c represent 95% confidence intervals.

Supplementary Fig. 2. Changes in SOC density (SOCD) towards higher latitudes in the north hemisphere (a) and south hemisphere (b).

Also, the extent of urban land has always been difficult to quantify. First, it changes rapidly. Second, there is not always agreement about what constitutes urban land. Third, metropolitan areas define their political boundaries in different ways (some cities include only the urban core, others annex rural land as it is developing). This also makes population counts in cities a rather arbitrary way to categorize them. Can the authors clarify this part of the methodology a bit more? Finally, patterns of urbanization take many different forms (see Schneider, A., & Woodcock, C. E. (2008). Compact, dispersed, fragmented, extensive? A comparison of urban growth in twenty-five global cities using remotely sensed data, pattern metrics and census information. *Urban Studies*, 45(3), 659-692.) The authors don't really address these challenges and do not make any estimates of how this might influence the uncertainty of results and in which direction. This topic merits considerably more discussion in the Uncertainties section.

Reply: We have now clearly defined urban built-up areas in the method section. “Urban

built-up areas were defined as regions dominated by continuous artificial impervious areas (>50%) or having a population density larger than 1500 habitants per km² regardless of political boundaries (Florczyk et al., 2019).” We understand that the urban areas can change over time and thus we have now clarified the reference year (i.e., 2015) of urban area data from GHSL Data Package (Florczyk *et al.*, 2019). We have now discussed that “In the context of continuing urbanization and urban greening (Chen *et al.*, 2020), total SOC stocks in global urban greenspaces will likely increase further over time in comparison to the current estimate (base year 2015)” Finally, we have also discussed the potential effects of urbanization forms that may lead to uncertainties of the estimates of urban areas.“Urbanization occurs in diverse patterns (e.g., compact, dispersed, fragmented, extensive) and this may made it challenging to accurately estimate the the areas of urban greenspaces using remotely sensed data (Schneider & Woodcock, 2008), potentially resulting in uncertainties of estimates for the national and global SOC stocks in urban greenspaces. Cities contain some greenspaces where vegetation is either planted (street trees) or overhang impervious surface, while the current upscaling approach may lead to an overestimate of surface SOC stocks in these greenspaces.”

References

- Chen, G. et al. Global projections of future urban land expansion under shared socioeconomic pathways. *Nat. Commun.* 11, 537 (2020).
- Florczyk AJ, Corbane C, Ehrlich D, Freire S, Kemper T, Maffenini L, Melchiorri M, Pesaresi M, Politis P, Schiavina M. 2019. GHSL data package 2019. *Luxembourg, EUR 29788(10.2760): 290498.*
- Schneider A., Woodcock, C. E. 2008. Compact, dispersed, fragmented, extensive? A comparison of urban growth in twenty-five global cities using remotely sensed data, pattern metrics and census information. *Urban Studies*, 45(3), 659-692.

Finally, I find it unconvincing that the global soil C baseline estimate, or the models that were used to derive it, provide any real useful guidance for soil management in

cities, other than to underline that soils should be managed in the first place and not simply scraped away (a practice that releases large amounts of carbon). I think the authors could clarify this (line 32).

Reply: Thanks and we have now revised the sentence as “Our findings present to our knowledge the first systematic assessment of surface-layer SOC in global urban greenspaces and provide a baseline for future urban soil C assessment.”

Significance

These are significant and important results that are essential for incorporating urban areas (where the majority of humans live) into various earth system models. This was a challenging project and will make a significant contribution to our understanding of the role of urban soils.

Reply: Thank you for the encouraging comments.

Data and methodology

While the supplemental files present the data used in this study, there doesn't seem to be any indication of the source of these data. I know numerous authors have compiled data from studies across the globe, but according to line 236-239, the authors did a literature search and compiled this data set. Perhaps I just missed it, but shouldn't the data set include the name of the study whence the data came? Without a source, there is no way to verify these data.

Reply: We have clearly described our approach to compile the SOC-U database in the revised method section. We have further updated the database and included references for data sources in Supplementary information. Now the whole database and corresponding references have been provided as supplementary data and will be formally archived in *Figshare* (<https://figshare.com/s/d03ab1150d644f2df301>) when our manuscript is accepted for publication. Please see more details in the Methods section and supplementary information.

Analytical approach

The analytical approach is sound, however, there are a few instances (already mentioned) where conclusions do not seem fully supported and where uncertainty is not fully described.

Reply: Thanks again for your suggestions. We have now updated the SOC-U database, conducted a reanalysis and revised the manuscript accordingly. Currently, the conclusions are fully supported by statistical results and potential uncertainties have been carefully discussed in the Section “Uncertainties and implications”. Also see our replies above.

Suggested improvements

Many of the suggestions below relate to topics described elsewhere in the review. They are presented in order of appearance in the manuscript, not in order of importance.

Reply: Thanks for your helpful suggestions. We have revised the manuscript accordingly.

Title and elsewhere: The authors use “distributions” in some cases in “distribution” in others. In this context, “distribution” makes more sense.

Reply: We revised it as “distribution” consistently.

L 21-23: This one of numerous instances where the comparison context is not clear. Is this total SOC? Or SOC per greenspace land area? Or SOC per city land area?

Reply: Sorry for any confusion. It should be “SOC density” in greenspace. We have revised it.

L 24: SOC density is usually expected to include the top 1 m of soil. These are only for 0-20 cm (from Table 1, presumably) but it needs to be clarified here or it will be misinterpreted.

Reply: We have now clearly mentioned the soil depth as “We mapped surface-layer SOC density in global urban greenspaces and estimated an average SOC density of

55.22 (51.86-58.59) (Mg C ha⁻¹) to the depth of 20 cm...". We have also clarified the soil depth where necessary in the revised manuscript to avoid potential misinterpretation.

L 27-30: The inclusion of "the top ten countries were all high income..." immediately before listing the U.S. SOC stocks gives the impression that this high SOC stock in the U.S. is associated with the large share of the world economy represented by the U.S. and conflates the density question with the total. This should be rephrased by perhaps saying "the ten countries with the highest SOC density were all high-income countries"). I would think it most likely that the U.S. has high level of SOC in urban greenspaces because it has low density, sprawling cities with low populations and lots of greenspace.

Reply: We have now removed the sentence "the top ten countries were all high income..." from the abstract. We now say that "The United States has the largest SOC stocks in urban greenspaces (0.37 Pg C) that accounts for one-fourth of the global total, while China (0.15 Pg C) and India (0.12 Pg C) jointly account for less than one-fifth of the global total.". Moreover, we have now mentioned in the discussion section that the U.S. has the largest SOC stock in urban greenspaces due to its largest area of urban greenspaces.

L 32: change "management" to "assessment". This study doesn't tell us much about management.

Reply: Done.

L 35-37: Urban land has been quantified as being up to 3% of global land. Can the authors qualify this a bit or give a range?

Reply: We have given a range of "Urban areas cover 0.3~ 0.6% of the world's land (Gao & O'Neill, 2020; Liu *et al.*, 2020)".

References

Gao, J, O'Neill BC. 2020. Mapping global urban land for the 21st century with data-driven simulations and shared socioeconomic pathways. *Nature*

Communications 11(1): 2302.

Liu X, Huang Y, Xu X, Li X, Li X, Ciais P, Lin P, Gong K, Ziegler AD, Chen A, et al. 2020. High-spatiotemporal-resolution mapping of global urban change from 1985 to 2015. *Nature Sustainability* 3(7): 564-570.

L 38: "...and are generally...source" could be deleted? Seems redundant with previous sentence.

Reply: Done as suggested.

L 41: largest C pool in comparison to what? This is unclear.

Reply: We have now removed the statements on "largest C pool".

L 54-59: The authors attempt to suggest that the urban convergence theory has been disproven by studies showing strong climate control of SOC. These aren't really contradictory as urban convergence is more about social norms driving design and management and the outside effects this sometimes has on plant communities, N, and water, thus causing cities to converge (not be the same, but move towards one another). Rephrase? Perhaps "However, especially at a broader scale, there is still strong..."

Reply: We have revised the sentence as suggested "At a much broader scale, recent observational analyses have suggested that there is nevertheless a strong latitudinal pattern and climate control of urban SOC density".

L 78-79: Delete? Seems like this point has already been made?

Reply: Done.

L 85 "urban built-up areas" isn't really a very helpful definition. Can the authors elaborate?

Reply: Urban built-up areas were defined as regions dominated by continuous artificial impervious areas (>50%) or having a population density larger than 1500 habitants per km² regardless of political boundaries (Florczyk et al., 2019). We have now clarified it

in the revised manuscript.

References

Florczyk AJ, Corbane C, Ehrlich D, Freire S, Kemper T, Maffenini L, Melchiorri M, Pesaresi M, Politis P, Schiavina M. 2019. GHSL data package 2019. Luxembourg, EUR 29788(10.2760): 290498.

L 90-94: the population cut-off is quite arbitrary. 500,000 is a small city in some regions and a large city in others. Can the authors justify this choice and explain the biases it might introduce? Can they indicate how the boundaries of these cities were determined (central city, metro area)?

Reply: The cut-off of populations for mid- and large cities (i.e., those with urban populations larger than 500,000) is defined by UN, Department of Economic and Social Affairs, Population Division (2015). It only accounts for the population living in the urban areas of each city. We have now clarified it in the revised manuscript. We also understand your concerns about the potential uncertainties for national estimates of SOC stocks. In fact, we scaled up to national estimates by including all urban areas. “To estimate the national SOC stocks in urban greenspaces, we first estimated the national urban greenspaces area ($UGSA_n$) according to Equation (5),

$$UGSA_n = UGSA_c \times \frac{UA_n}{UA_c} \quad (5)$$

where $UGSA_c$ is the total urban greenspaces area of middle- and large cities (ha) in a country (derived from Huang *et al.*, 2021), UA_c is the corresponding total urban built-up area (ha) of these cities (derived from Florczyk *et al.*, 2019), and UA_n is the total urban built-up area of all cities in that country (derived from Florczyk *et al.*, 2019). The national total SOC stocks of urban greenspaces ($SOCS_n$, Tg C) was estimated as Equation (6),

$$SOCS_n = SOCS_c \times \frac{UGSA_n}{UGSA_c} \quad (6)$$

where $SOCS_c$ is the sum of the SOC stocks of middle- and large city (ha) in that country (The national-level SOC stocks in urban greenspaces was given in supplementary data 2).” Thanks for your understanding.

References

- Florczyk AJ, Corbane C, Ehrlich D, Freire S, Kemper T, Maffenini L, Melchiorri M, Pesaresi M, Politis P, Schiavina M. 2019. GHSL data package 2019. *Luxembourg, EUR 29788(10.2760): 290498.*
- Huang C, Yang J, Clinton N, Yu L, Huang H, Dronova I, Jin J. 2021. Mapping the maximum extents of urban green spaces in 1039 cities using dense satellite images. *Environmental Research Letters* 16(6): 064072.
- UN, Department of Economic and Social Affairs, Population Division. 2015. World urbanization prospects: The 2014 revision. *United Nations Department of Economics and Social Affairs, Population Division: New York, NY, USA* 41.

L 97-106: This is a context issue that occurs in several places. Are these figures for density for cities? (as stated on line 100) or for urban greenspaces within cities?

Reply: Sorry for any confusion. These figures are for urban greenspaces. We have now revised it as “Observed values of surface SOC concentrations (0-20 cm) in urban greenspaces varied considerably across cities (Supplementary Fig. 1a and Supplementary Table 4), ranging from 3.6 to 101.0 g kg⁻¹ with a global geometric mean of 24.6 g kg⁻¹ (median = 25.1 g kg⁻¹). Surface SOC density of urban greenspaces also varied greatly across cities (Fig. 1b), ranging from 8.8 to 112.0 Mg C ha⁻¹ with a global geometric mean of 51.4 Mg C ha⁻¹ (median = 57.0 Mg C ha⁻¹).”.

L 110: I was surprised at the inclusion of soil depth here. Soil depth in cities is often related to construction practices and prior land use. It is notoriously erratic. Individual studies tend to define depth in relation to the questions of interest for that study and are thus inconsistent. Where did depth data come from and how was it defined?

Reply: We have now included more detailed criteria of data collection in the revised method section. Specifically, we clarified that “... (iii) soil samples were collected within the top 20 cm depth (as reported in original literature)”. We further mentioned that “We only considered the surface soil layer because (i) surface soils are strongly affected by vegetation and human activities, (ii) surface soils are essential for nutrient

retention and supply for plant growth, and (iii) surface SOC data are more available in literature. Reported data for replicated samples of a same vegetation type within a same sampling site were averaged for further analysis.”

L121: The authors make much of the role of GDPP on SOC density here and elsewhere. I do not find this conclusion convincing. First GDPP only explains 8.7% of the variance. Second, I am not convinced it is not correlated with latitude. In L 128 the authors say there is low multicollinearity of variables, but it is difficult for me to see how latitude and GDPP can be separated based on the cities represented in the data. Many of the top ten cities are presumably located on what was previously boreal forest, the forest cover with the highest SOC density in the world. Why did the authors not compare SOC density with the SOC density of surrounding nonurban land?

Reply: We fully understand your concern about the role of GDPP on SOC density. In our revised manuscript, we have updated the database (now including 420 observations from 257 cities in 52 countries) and the sample size has increased substantially in our updated database (420 vs. 161). We further conducted a reanalysis by including more explanatory variables, such as vegetation type, temperature seasonality, and precipitation seasonality. Different from the previous results based on smaller sample size, our re-analyses based on linear models and random forest models both show that GDPP is not an important predictor for global patterns of topsoil SOC density in urban greenspaces (Fig. 2a & 2b, also see below). This implies that our previous results might be biased. We have now revised the manuscript accordingly, i.e., “Other potential factors (i.e., MAP, vegetation type, UHI, GDPP, PD, and elevation), however, showed lower importance in explaining the global patterns of SOC density in urban greenspaces (Fig. 2a & 2b).” Additionally, we were not able to compare SOC density in urban greenspaces with the SOC density of surrounding nonurban land due to a lack of paired data on nonurban land. Thanks for your understanding.

Fig. 2. Estimated importance of explanatory variables based on model selection analysis (sum of the Akaike weights for the linear mixed-effects models in which the explanatory variable appeared) (a) and random forest model analysis (Mean Decrease Gini) (b). See the revised method section for more details.

L 184-5: I'm not really sure the "better" vs "poor" management are appropriate terms. More irrigation or N additions may result in greater SOC density, but is it really "better" management? I'm also not sure a broad generalization about developing countries can be made from the Zhang et al. 2022 paper. In addition, papers regarding SOC accumulation over time are looking at small time scales (say 30 years) and are not really applicable to comparisons between ancient urban cores and newer development where the time scale could be 100s or 1000s of years. I think the authors need to rethink some of the generalizations here.

Reply: Thanks for your insightful comments. We agree with your view on the management operations and we have now removed previous statements from the revised manuscript. Based on updated results, we have now discussed that "Surprisingly, we found that anthropogenic variables (e.g., UHI, GDPP and PD) and vegetation type exerted limited effect on the spatial pattern of SOC density on a global scale. In contrast, anthropogenic drivers are more likely to affect soil SOC locally. For example, management operations in urban greenspaces (e.g., selection of plant species for urban

greening, nutrient fertilization, irrigation, and pest control) can favour vegetation growth and SOC accumulation (Dobbs *et al.*, 2014), but such effects may be unable to substantially alter the global pattern of SOC density.”

We have now also discussed the potential roles of the age of urban greenspaces. We have now mentioned that “The ages of greenspaces are indicative for the length of time to accumulate SOC as topsoil organic matters are often stripped away and lost during the construction of greenspaces, but we were unable to derive information on the ages of urban greenspaces for a quantitative analysis.”

Reference

Dobbs C, Nitschke CR, Kendal D. 2014. Global drivers and tradeoffs of three urban vegetation ecosystem services. *PLoS One* 9(11): e113000.

L 211-227: Additional uncertainties and their impacts should be included here. These include potential mismatches between the data sets used for the SOC vs the cities studied, questions of landuse history, etc. The authors rightly point to high levels of HAHT soils in place in cities. This trend is in fact increasing in many cities and the potential for C sequestration in some of these soils is unknown and likely quite different from traditional soils. In addition, many soils with high organic matter content are buried (buried A horizons) in some cities. Can the authors provide some estimates on the uncertainty? What was the range of SOC densities reported from the actual data vs predicted results from the models. Perhaps I missed it, but I did not see this kind of analysis in this study.

Reply: We agree with the comment that anthropogenic disturbances, such as land use history, topsoil removal and/or import of soils from elsewhere, may made lead to uncertainties in our global mapping of SOC density in urban greenspaces. We have now discussed this in the revised manuscript that “Anthropogenic disturbances, such as land use change, topsoil removal, topsoil bury and/or import of soils from elsewhere, are common during urban expansion, but we were not able to evaluate their potential effects in our analysis.”

Regarding the analysis of uncertainty, we first evaluated the model performance using determination coefficient (R^2) and root mean square error (RMSE) (Supplementary Fig. 6, also see below). The reanalysis using the updated database showed that random forest model had better performance than the best linear models. We further evaluated the relative uncertainty of the predicted SOC density using the coefficient of variance (C.V., %) of ten simulations when training the random forest models (Supplementary Fig. 8, also see below). The results indicate that our global prediction of SOC density in urban greenspaces had relatively low levels of uncertainty (the coefficient of variance of simulations mostly $< 5\%$). Moreover, uncertainties of mean values were reported as the 95% confidence interval in the revised manuscript.

Supplementary Fig. 6. Density plot comparing cross-validated RMSE (a) and cross-validated R^2 for the linear model (LM) and the random-forest model (RF).

Supplementary Fig. 8. Coefficients of variance (CV, %) of predicted surface-layer (0-20 cm) soil organic carbon density (SOCD) in global urban greenspaces.

Fig 3 b and d: b shows SOC density, while d shows total stocks. I think it would be informative to have some indication of density by country or also perhaps proportion of urban land represented by a given country. Just seeing how much total is in each country isn't as informative.

Reply: Thanks for your suggestion. We have now updated the figure by including country-level urban greenspace area and illustrate the SOC stocks in a separate figure (Fig. 3 for SOCD and Fig. 4 for SOC stocks, also see below).

Fig. 3 Global patterns of predicted surface SOC density (SOC) (0-20 cm) and area-weighted national mean SOC density (SOCD) in urban greenspaces. **a**, Predicted SOCD of urban greenspaces for mid- and large cities (urban population > 0.5 million) (Supplementary data 1). **b**, Average SOCD of urban greenspaces estimated for the globe and top-10 countries weighted by national areas. Error bars indicate the 95th and 5th percentiles.

Fig. 4 Global patterns of surface SOC stocks (SOCS) (0-20 cm) of mid- and large cities and national estimates of SOCS in urban greenspaces. **a**, Predicted SOCS of urban greenspaces for mid- and large cities (urban population > 0.5 million) (Supplementary data 1). **b-c**, Total SOCS (**b**) and urban greenspace areas (UGSA) (**c**) estimated for the globe and top-10 countries. The estimates of national SOCS were based on the total national areas of urban greenspaces (Supplementary data 2). Error bars indicate the 95th and 5th percentiles.

Clarity and context

In general, the paper is clearly presented. However, there are numerous locations where the scale or context is not clear, especially when comparisons are being made. While it can likely be puzzled out, a few additional words would be helpful to readers. These are noted in the Suggested improvements section.

Reply: We have now thoroughly revised the manuscript and improved the clarity. See more details in our reply to your comments above and in the revised manuscript.

References

References seem appropriate, except as noted elsewhere.

Reply: Thanks again. We have checked and updated the references.

REVIEWERS' COMMENTS

Reviewer #1 (Remarks to the Author):

Thank you for the response to my review. I am pleased to see that the revisions made have yielded a broader, more replicable dataset and a number of important clarifications have been made. I feel the study is much improved and appreciate the time the authors put into these improvements and the response.

My response below related to the original numbering of my comments:

Comments 1. and 2.

* I was surprised to learn that the previous manuscript focused on urban forests only. The expansion to other urban greenspace types is indeed valuable and necessary for the manuscript to claim to provide an analysis of urban greenspaces. Hence, I agree this is a good addition.

* I am very happy to see an updated database. This looks well organised and has helped to significantly improve the study overall. I am sure it will be useful to others in the community.

* Thank you for including the search string and details of where the searches were made. This is quite a short search string for a study of this nature. For example, garden isn't included or alternative words for 'urban'. Did you search for these terms in titles/abstracts/keywords? I also assume that you used syntax in the search that allowed it to deal with plurals or alternative word endings? I am not suggesting that you expand/improve the search at this stage, however, this is the level of detail needed to call this a systematic review. If it was a systematic review, I would expect to see a more robust search string, details on the number of articles returned, details of a process for screening and data abstraction, all following an accepted protocol for systemic review such as PRISMA. Hence, I suggest that you remove the term systematic from the manuscript to avoid misleading the reader.

* Thank you for including further detail regarding how SOC density was calculated. I also appreciate the inclusion of the test of the method for estimated bulk density requested by the other reviewer.

Comment 3. Thank you for these edits, this is clearer now.

Comment 4. I see, thank you for the explanation.

Comment 5. This is a useful addition. I am a little surprised that it is the least important predictor. I wonder whether this is because the vegetation type is not well recorded - given so many records without a vegetation type and different types of trees being gathered together as urban forest? I suggest it may be worth noting that improved information on plant functional type/habitat type of sampling locations may be needed to understand the importance of that variable in the limitations section, with reference to studies such as Weismeier et al 2019 (<https://doi.org/10.1016/j.geoderma.2018.07.026>). Along similar lines, in the discussion around line 204 you mention that it is surprising that anthropogenic factors do not play a greater role. I would argue perhaps that's because you don't have the sufficient resolution of data or more direct factors relating to management. It also leads me to wonder whether it

is the scale of analysis that leads to climate being the predominant factors - as highlighted by a study by Nave et al 2021 (<https://doi.org/10.1007/s10533-020-00745-9>). I leave it to the authors to consider if this changes any of the discussion on predictors.

Comment 6. Thank you for being open to look at this, and I am glad it's been a useful exercise.

Comment 7. I agree that the analysis has improved and appreciate the inclusion of a limitations section.

Comment 8. Thank you for this.

Comment 9. Interesting to see that now GDPP is a lesser factor than regions such as China and India still have lower SOCD, and I agree that in your model this will likely be related to the climate variables. In the response I don't see how the age of development factors into your results? You don't have this as a predictor, as you point out there isn't sufficient data to support that. Are you perhaps suggesting that some of the effects attributed to MAT and seasonality might in fact also be related to a co-variate of age of development as urban areas in mid latitudes are more likely to be more recently disturbed and thus may have lower SOCs? If this is being suggested in the response, it also needs to be included in the manuscript.

10. That's fine, thanks.

11. This is clearer now thank you

Reviewer #3 (Remarks to the Author):

The authors have addressed all of my concerns.

REVIEWERS' COMMENTS

Reviewer #1 (Remarks to the Author):

Thank you for the response to my review. I am pleased to see that the revisions made have yielded a broader, more replicable dataset and a number of important clarifications have been made. I feel the study is much improved and appreciate the time the authors put into these improvements and the response.

Reply: Many thanks for your helpful comments to improve our work. We are pleased that you are now satisfied with our revised manuscript.

My response below related to the original numbering of my comments:

Comments 1. and 2.

* I was surprised to learn that the previous manuscript focused on urban forests only. The expansion to other urban greenspace types is indeed valuable and necessary for the manuscript to claim to provide an analysis of urban greenspaces. Hence, I agree this is a good addition.

* I am very happy to see an updated database. This looks well organised and has helped to significantly improve the study overall. I am sure it will be useful to others in the community.

* Thank you for including the search string and details of where the searches were made. This is quite a short search string for a study of this nature. For example, garden isn't included or alternative words for 'urban'. Did you search for these terms in titles/abstracts/keywords? I also assume that you used syntax in the search that allowed it to deal with plurals or alternative word endings? I am not suggesting that you expand/improve the search at this stage, however, this is the level of detail needed to call this a systematic review. If it was a systematic review, I would expect to see a more robust search string, details on the number of articles returned, details of a process for screening and data abstraction, all following an accepted protocol for systemic review such as PRISMA. Hence, I suggest that you remove the term systematic from the manuscript to avoid misleading the reader.

* Thank you for including further detail regarding how SOC density was calculated. I also appreciate the inclusion of the test of the method for estimated bulk density requested by the other reviewer.

Reply: Thanks for your comments and suggestions. We have now removed the term “systematic” to ensure the accuracy and clarity of descriptions according to your suggestions.

Comment 3. Thank you for these edits, this is clearer now.

Reply: Thanks.

Comment 4. I see, thank you for the explanation.

Reply: Thanks.

Comment 5. This is a useful addition. I am a little surprised that it is the least important predictor. I wonder whether this is because the vegetation type is not well recorded - given so many records without a vegetation type and different types of trees being gathered together as urban forest? I suggest it may be worth noting that improved information on plant functional type/habitat type of sampling locations may be needed to understand the importance of that variable in the limitations section, with reference to studies such as Weismeier et al 2019 (<https://doi.org/10.1016/j.geoderma.2018.07.026>). Along similar lines, in the discussion around line 204 you mention that it is surprising that anthropogenic factors do not play a greater role. I would argue perhaps that's because you don't have the sufficient resolution of data or more direct factors relating to management. It also leads me to wonder whether it is the scale of analysis that leads to climate being the predominant factors - as highlighted by a study by Nave et al 2021 (<https://doi.org/10.1007/s10533-020-00745-9>). I leave it to the authors to consider if this changes any of the discussion on predictors.

Reply: Thanks for your insightful comments. We have now included additional discussions on the vegetation type and anthropogenic factors following your suggestions.

First, we agree with your comments on the uncertainties in our analysis of vegetation type possibly due to incomplete data recording. In our recent revision of the manuscript, we have tried our best to add the information on urban greenspace types based on descriptions in the original literature. Unfortunately, the updated database is still limited to some data gaps. In our revised manuscript, we have now mentioned it as *“The low importance of vegetation type could possibly be attributable to a coarse classification (e.g., urban forest and urban lawn) and a potentially masking effect of climate variables on the role of vegetation.”*

Second, we agree with your comments that the resolution and availability of anthropogenic data also limit our analysis of the role of anthropogenic factors. We have now discussed this limitation as *“Moreover, high nitrogen deposition and rising levels of atmospheric CO₂ in urban environments often favour plant growth and enhance SOC accumulation (Pouyat & Trammell, 2019; Du et al., 2022; Wang et al., 2023). However, we were unable to conduct a quantitative analysis to incorporate such an “urban hotspot effect” (Du et al., 2016), again due to the lack of high-resolution data within cities. Anthropogenic disturbances, such as land use change, topsoil removal, and/or import of soils*

from elsewhere, are common during urban expansion, but we were not able to evaluate their potential effects in our analysis. The limitation of the above-mentioned data could lead to a potential underestimation of the influence of anthropogenic factors on the spatial variations of SOC in urban greenspaces.”

Third, we concur that the key drivers of SOC density likely vary across spatial scales. We have now discussed it as “The observed predominance of climatic variables over anthropogenic variables in shaping the global patterns of urban SOC density suggests that the key drivers of SOC density likely vary across spatial scales. Climatic drivers have been found to determine spatial variations of SOC at a continental or global scale (Weismeier et al., 2019; Nave et al., 2021). In contrast, anthropogenic drivers are likely more influential to affect SOC locally. For example, management operations in urban greenspaces (e.g., selection of plant species for urban greening, nutrient fertilization, irrigation, and pest control) can favour vegetation growth and SOC accumulation (Dobbs et al., 2014), but such effects may be unable to substantially alter the global pattern of SOC density.”

References:

- Pouyat, R. V. & Trammell, T. L. in: Global Change and Forest Soils (eds Busse, M. et al.) 189-211 (Elsevier, 2019).
- Du, E. et al. Ecological effects of nitrogen deposition on urban forests: an overview. *Front. Agric. Sci. Eng.* 9, 445-456 (2022).
- Wang, Y. et al. Urban CO₂ imprints on carbon isotope and growth of Chinese pine in the Beijing metropolitan region. *Sci. Total Environ.* 866, 161389 (2023).
- Du, E. et al. Imbalanced phosphorus and nitrogen deposition in China's forests. *Atmos. Chem. Phys.* 16, 8571-8579 (2016).
- Wiesmeier, M. et al. Soil organic carbon storage as a key function of soils - A review of drivers and indicators at various scales. *Geoderma.* 333, 149-162 (2019).
- Nave, LE. et al. Patterns and predictors of soil organic carbon storage across a continental-scale network. *Biogeochemistry.* 1-22 (2021).
- Dobbs, C., Nitschke, C. & Kendal, D. Global drivers and tradeoffs of three urban vegetation ecosystem services. *PLoS One.* 9, (2014).

Comment 6. Thank you for being open to look at this, and I am glad it's been a useful exercise.

Reply: Thanks.

Comment 7. I agree that the analysis has improved and appreciate the inclusion of a limitations section.

Reply: Thanks.

Comment 8. Thank you for this.

Reply: Thanks.

Comment 9. Interesting to see that now GDPP is a lesser factor that regions such as China and India still have lower SOCD, and I agree that in your model this will likely be related to the climate variables. In the response I don't see how the age of development factors into your results? You don't have this as a predictor, as you point out there isn't sufficient data to support that. Are you perhaps suggesting that some of the effects attributed to MAT and seasonality might in fact also be related to a co-variate of age of development as urban areas in mid latitudes are more likely to be more recently disturbed and thus may have lower SOCs? If this is being suggested in the response, it also needs to be included in the manuscript.

Reply: Thanks for your insightful comments and suggestions. We agree with your comments that the age of urban greenspaces can be an important predictor of SOC density as it is indicative for the length of time to accumulate SOC. However, we were unable to derive information on the ages of urban greenspaces for a quantitative analysis. We have included additional discussion as suggested by the reviewer. We now highlight the role of the ages of urban greenspaces that *“higher SOC density often occurs in older urban greenspaces (Huyler et al, 2014; Scharenbroch et al., 2017).”* We further discussed that *“Given that a large proportion of newly established urban areas and greenspaces are located in developing countries at mid- to low latitudes (Sun et al., 2020), it is possible that the ages of these urban greenspaces correlate with MAT and temperature seasonality. This potential collinearity might result in an overestimation of the effect of MAT and temperature seasonality on the spatial pattern of SOC density in our analysis. Unfortunately, our ability to conduct a quantitative assessment of the ages of urban greenspaces was constrained by the limited availability of relevant data.”* Thanks for your understanding.

References:

Huyler, A., Chappelka, AH., Prior, SA. & Somers, GL. Drivers of soil carbon in residential ‘pure lawns’ in Auburn, Alabama. *Urban Ecosyst.* 17, 205-219 (2013).

Scharenbroch, B., Day, S., Trammell, T. & Pouyat, R. in: Urban Soils (eds Lal, R., Stewart, BA.)
137-154 (CRC Press, 2017).

Sun, L., Chen, J., Li, Q. & Huang, D. Dramatic uneven urbanization of large cities throughout the
world in recent decades. Nat. Commun. 11, 5366 (2020).

10. That's fine, thanks.

Reply: Thanks.

11. This is clearer now thank you

Reply: Thanks.

Reviewer #3 (Remarks to the Author):

The authors have addressed all of my concerns.

Reply: Many thanks for your helpful comments to improve our work. We are pleased that you are
now satisfied with our revised manuscript.